# Streaming Federated Learning with Markovian Data

**Tan-Khiem Huynh**[*]
Inria,
INSA Lyon

**Malcolm Egan**
Inria,
INSA Lyon

**Giovanni Neglia**
Inria,
Université Côte d'Azur

**Jean-Marie Gorce**
Inria,
INSA Lyon

## Abstract

Federated learning (FL) is now recognized as a key framework for communication-efficient collaborative learning. Most theoretical and empirical studies, however, rely on the assumption that clients have access to pre-collected data sets, with limited investigation into scenarios where clients continuously collect data. In many real-world applications, particularly when data is generated by physical or biological processes, client data streams are often modeled by non-stationary Markov processes. Unlike standard i.i.d. sampling, the performance of FL with Markovian data streams remains poorly understood due to the statistical dependencies between client samples over time. In this paper, we investigate whether FL can still support collaborative learning with Markovian data streams. Specifically, we analyze the performance of Minibatch SGD, Local SGD, and a variant of Local SGD with momentum. We answer affirmatively under standard assumptions and smooth non-convex client objectives: the sample complexity is proportional to the inverse of the number of clients, with a communication complexity comparable to the i.i.d. scenario. However, the sample complexity for Markovian data streams remains higher than for i.i.d. sampling. Our analysis is validated via experiments with real pollution monitoring time series data.

## 1 Introduction

As edge networks grow in scale, clients are increasingly diverse [34]. Many of these clients are capable of continuously collecting and learning from data, instead of simply storing large quantities of historical data [28]. This *streaming data* arises in applications ranging from health [21, 22] and environmental monitoring [24], to control of robots [29].

Each data stream provides detailed information about individual clients. However, applications such as environmental resource management [54] or public health monitoring [6] often aim to derive population-level insights, leveraging data streams from multiple individuals to learn a global model. As this data may be sensitive or costly to communicate, centralized training solutions are undesirable. Federated Learning (FL) has emerged as a key strategy for communication-efficient collaborative learning [26]. In conventional FL scenarios, clients have access to local datasets used to train models locally. These models are then transmitted to a central server, where aggregation facilitates collaborative learning. FL minimizes communication overhead while learning models with high accuracy and ensuring client data are not shared.

Clients with streaming data must also cope with memory constraints [38, 35]. This means that clients have limited control over data sampling: local updates can only be computed using the current data

---

[*]Inria, INSA Lyon, CITI, UR3720, 69621 Villeurbanne, France

39th Conference on Neural Information Processing Systems (NeurIPS 2025).

samples stored in memory. Moreover, data in health and environmental monitoring applications is often generated by non-linear physical and biological systems. The data generated by these systems is frequently modeled as non-stationary Markovian data streams [1, 58]. When client memory caches are continually refreshed by Markovian streaming data, the samples available to compute updates are statistically dependent and non-stationary *in time*.

In this paper, we address the following question: *Can FL with Markovian data streams support collaborative learning?* While Markovian sampling has been explored in the centralized setting [17], existing work in the federated setting has focused on Federated Reinforcement Learning [45, 28, 37] with convex objectives. However, classification and regression tasks in systems with non-linear dynamics often rely on deep neural networks or non-convex regularization [63, 36], resulting in non-convex minimization problems during training. We therefore consider a family of training tasks beyond the work in [45, 28, 37] characterized by *smooth non-convex loss functions*. Our main contributions can be summarized as follows:

   (i) *Impact of Markovian sampling:* We rigorously characterize the impact of Markovian sampling on the convergence rate of Minibatch and Local Stochastic Gradient Descent (SGD), the two most common baselines in FL. A key conclusion is that linear speed-up is achieved under standard assumptions on the objectives and heterogeneity: the sample complexity is proportional to the inverse of the number of clients.

  (ii) *Mitigating client heterogeneity:* We show that a momentum-based variant of Local SGD introduced in the i.i.d. setting [9, 61] can effectively mitigate the client drift with Markovian data streams. In particular, unlike Local SGD, Local SGD with Momentum matches the performance of Minibatch SGD up to a constant without requiring any bound on client data heterogeneity.

 (iii) *Validation on Environmental Monitoring Data:* The key difference between i.i.d. and Markovian sampling is statistical dependence between consecutive samples. We validate our analysis using multi-site pollution time series data [7], which shows the benefit of collaboration on real data with dependent samples.

## 2   Related Works

Edge devices often encounter the challenge of processing continuous data streams while operating under memory constraints [38, 35]. Learning from streaming data presents numerous challenges, with solutions dependent on the data-generating process. One line of research addresses scenarios where the data distribution evolves over time, with continual learning methods designed to adapt the model to the changing data distribution [51]. Another line focuses on adversarial settings, where the data stream is generated by an adversary [53], often addressed using online learning algorithms with regret guarantees.

In this paper, we focus on learning from data generated by Markovian processes. Markovian processes have been successfully applied to model various physical and biological systems [1, 58], and encompass the special case where data samples are drawn i.i.d. from an unknown distribution.

In the centralized setting, this learning problem can be framed as a stochastic approximation problem, typically addressed by variants of stochastic gradient descent (SGD). While SGD with i.i.d. data is well-understood [47, 31], the Markovian nature of the data stream introduces challenges due to the temporal correlation between samples.

For uniformly ergodic Markov chains, [16] first demonstrated that mirror descent achieves optimal convergence rates for Lipschitz, general convex, and non-convex problems. Recently, [15] proposed a random batch size algorithm that adapts to the mixing time of the Markov chain, while [3] adopted the same technique to develop accelerated methods and established lower bounds for strongly convex objectives.

For general Markov chains, existing analysis of SGD-type algorithms [55, 14, 13] have shown sub-optimal dependence on the mixing time and on the variance of stochastic gradient estimates relative to the lower bounds in [3, 16]. We refer to [3, Table 1] for an exhaustive review of the literature on SGD with Markovian noise.

In the case of SGD with Markovian noise on a finite state space—a common scenario in distributed optimization [4]—[17, 23] developed convergence theories and variance reduction techniques for first-order methods, demonstrating improvements over the gossip algorithm [5].

In the context of FL, a few papers have considered learning from data streams. [41] and [62] framed the problem in an adversarial and in a continual learning setting, respectively. [8] imposed strong homogeneity assumptions, such as requiring all clients to share a common optimal model. [38] assumed that each client's data stream consists of i.i.d. samples, while allowing for heterogeneous data distributions across clients. The study focused on temporal dependencies arising from the use of local memory and analyzed the impact of heterogeneous memory constraints. [35] studied how to jointly tune batch sizes and number of local updates, but did not have convergence results in the streaming setting.

In the case of Markovian data streams, recent work has primarily focused on Federated Reinforcement Learning (FRL) with *linear function approximation*. A key challenge is demonstrating that collaboration is beneficial in this setting; namely, that the per-client sample complexity of FRL algorithms decreases inversely with the number of clients. While this *linear speed-up* is well-understood for i.i.d. data, the impact of collaboration remains an open question in the presence of statistical dependence, as is the case with Markovian data streams. [28, 18] demonstrated the benefits of cooperation under a strong homogeneity assumption on the client environment. Heterogeneous settings have been considered in [12, 25] with a linear speed-up established in [59] subject to heterogeneity assumptions, which were then relaxed in [37] by incorporating control variates [27]. Unfortunately, these results do not readily extend beyond the FRL setting.

Beyond Markovian data streams, temporal correlations can arise from other factors in the FL setting. For instance, memory update strategies, as explored in [38], can introduce dependencies between subsequent updates. Similarly, the availability of clients can exhibit Markovian behavior, as discussed in [46, 49, 56]. While these sources of correlation are distinct from the Markovian data stream focus of this paper, they highlight the broader significance of understanding and mitigating temporal dependencies in the FL context.

# 3 Problem Setup

In this section, we introduce our problem setting. We denote by $[M] \coloneqq [1, M]$ the set of positive integers up to $M$, and by $[K]_0 \coloneqq [0, K-1]$ the set of non-negative integers less than $K$.

## 3.1 Streaming Federated Learning

Consider $M$ clients, each with an objective given by

$$F_m(w) = \mathbb{E}_{x \sim \pi_m}[f_m(w; x)],$$

where $f_m : \mathbb{R}^d \times \Omega_m \to \mathbb{R}$ and $\pi_m$ is the target data distribution. for client $m$. In the context of supervised learning, the objective $F_m$ corresponds to the true risk of the loss function $f_m$, parameterized by the parameter $w$, on a data sample $x$ drawn from $\pi_m$.

In FL, the $M$ clients collaborate to solve

$$\min_{w \in \mathbb{R}^d} \frac{1}{M} \sum_{m=1}^{M} F_m(w), \tag{1}$$

by iterating a two-phase procedure over multiple *communication rounds*. Two popular examples of this procedure are Minibatch SGD (Algorithm 2 in Appendix A) and Local SGD or FedAvg (Algorithm 3 in Appendix A) [26]. In the first phase of the $t$-th communication round, client $m$ has access to $K$ data samples. These samples are used to compute a local update: either a gradient estimate $g_t^{(m)}$ in the case of Minibatch SGD, or a model iterate $w_t^{(m,K)}$ obtained after performing $K$ local gradient steps in the case of Local SGD. In the second phase, the server aggregates the local updates from all clients—typically via averaging—to update the global model. In the second phase, the updates are aggregated via averaging.

In the standard FL scenario, the samples available at client $m$ in any communication round $t$ are fixed, corresponding to pre-collected data [26]. In contrast, in streaming FL, the data available to client

$m$ can change in every communication round. Streaming FL models scenarios where clients have a limited memory cache of $K$ samples and new data is collected over time [38, 35].

Streaming FL raises a key challenge: the data sampling process is not completely controlled by the clients. As a consequence, i.i.d. data samples from the target distribution $\pi_m$ may not be available. For example, patient data in health monitoring is continuously collected from wearable devices. Statistical dependence between data collected over time naturally arises due to latent factors such as daily routines, sleep patterns, and weather conditions [32]. Moreover, the initial data collection or warm-up period [2] might capture atypical behavior, such as adjustments to new wearable devices. As a consequence, the data stream of a patient $m$ will, in general, be non-stationary and not reflect the long-term target distribution $\pi_m$.

## 3.2 Markovian Data Streams

Data streams arising in health and environmental monitoring often exhibit specific statistical structures. Indeed, many biological, chemical, and physical systems are governed by non-stationary Markov processes [1, 58]. Data streams arising in federated reinforcement learning also form Markov processes [28, 44]; however, the use of stationary exploration policies leads to stationary data streams.

We model the data stream $X_m = (x_t^m)_{t \in \mathbb{N}}$ of client $m$ by a time-homogeneous Markov chain evolving on the state space $\Omega_m \subseteq \mathbb{R}^d$ with the corresponding Borel $\sigma$-field $\mathcal{B}_m$ [48, 39]. As an abuse of notation, for all $t \in \mathbb{N}$, and for all $k \in [K]_0$, we denote by $x_t^{(m,k)} := x_{Kt+k}^m$ the $k$-th sample available to client $m$ at the $t$-th communication round. We may also use $x_t^m := \left( x_t^{(m,k)} \right)_{k \in [K]_0}$ to denote all $K$ data samples available to client $m$ in communication round $t$. The evolution of the time-homogeneous Markov chain for client $m$ is characterized by the initial distribution $x_0^{(m,0)} \sim \mu_m$ and by the transition kernel $P_m$.

We focus on the case of independent clients, where $X_m$ is independent of $X_{m'}$ for $m \neq m'$. This scenario occurs when client data streams are generated by non-interacting processes. Key examples include patient health monitoring and environmental monitoring across spatially separated regions, where observations from different clients are often assumed to be independent [50].[2]

The Markov chain $X_m$ admits a stationary distribution $\pi_m$ on $\Omega_m$ if

$$\int_{x \in \Omega_m} \mathrm{d}\pi_m(x) P_m(x, \mathrm{d}x_t^{(m,k)}) = \mathrm{d}\pi_m(x_t^{(m,k)}).$$

We say that $X_m$ is stationary if $x_0^{(m,0)} \sim \pi_m$; otherwise, $X_m$ is non-stationary.

The stationary distributions $\pi_m$, $m \in [M]$ correspond to the target distributions in (1), which capture the long-term statistics of the data samples. Since long-term statistics, rather than transient behavior of the samples, are of interest for learning in health and environmental monitoring scenarios, the stationary distribution is a natural choice for the target distribution. Hence, we make the following assumption, which guarantees that the Markov process associated with each client admits a unique stationary distribution.

**Assumption 3.1.** The data samples of client $m \in [M]$ are drawn from an independent time-homogeneous Markov chain $X_m$ defined on $(\Omega_m, \mathcal{B}_m)$ with transition kernel $P_m$ and initial distribution $\mu_m$, converging to the unique stationary distribution $\pi_m$.

Assumption 3.1 ensures that samples $x_t^m$ from the data stream of client $m$ will be approximately drawn from $\pi_m$ as $t$ diverges.

We also define by $X = (X_m)_{m \in [M]}$ the *system-level Markov process* defined on $(\Omega, \mathcal{B})$, where $\Omega := \bigtimes_{m=1}^M \Omega_m$ and $\mathcal{B} := \bigotimes_{m=1}^M \mathcal{B}_m$. This Markov chain, at each time step, evolves independently on each of $M$ coordinates according to the corresponding transition kernel $P_m$. We denote by $P$ its transition matrix. Similarly, for the ease of notation, we write $x_t^k := \left( x_t^{(m,k)} \right)_{m \in [M]}$ and $x_t := \left( x_t^k \right)_{k \in [K]_0}$.

---

[2]In the case of nearby or overlapping regions, statistical dependence between the data samples may arise [40].

In Proposition B.6, we provide a formal characterization of $P$ and we also prove that $X$ is indeed a well-defined Markov chain. Furthermore, under Assumption 3.1, $X$ admits a stationary distribution $\pi = \bigotimes_{m=1}^{M} \pi_m$. We denote by $\nu_{ps}$ the *pseudo spectral gap* of $P$ [43, Section 3.1].

We also make the following assumption on the absolute continuity of the transition kernel $P$ with respect to the stationary distribution $\pi$.

**Assumption 3.2.** For every $x \in \Omega$, the probability measure $P(x, .)$ is absolutely continuous with respect to the stationary measure $\pi$, and its Radon-Nikodym derivative $\frac{dP(x, .)}{d\pi}$ is uniformly bounded on sets with non-zero measure with respect to $\pi$. In particular,

$$C_\infty = \sup_{x \in \Omega} \operatorname{ess\,sup} \left| \frac{dP(x, .)}{d\pi} \right| < \infty$$

We note that Assumption 3.2 is generally mild, in the sense that it only requires, for every $x \in \Omega$, $P(x, .)$ is absolutely continuous with respect to $\pi$, and every $\pi$-integrable function is also $P(x, .)$-integrable. This assumption is commonly used in the analysis of other stochastic approximation algorithms with Markovian noise, for example, Markov chain Monte Carlo [19, Theorem 12]. In the case where $\Omega$ is finite, $C_\infty = \max_{x,y \in \Omega} \frac{P(x,y)}{\pi(y)}$. We provide further discussion of the dependence of $C_\infty$ on the transition kernel in Appendix D.5.

In contrast to most existing studies on stochastic optimization (approximation) with Markovian data, we do not impose any assumption on the speed of convergence to the stationary measure of the underlying Markov processes, such as the commonly used uniform geometric ergodicity assumption (see, e.g., [3, Assumption A3; 37, Assumption A3; 59, Assumption 3; 65, Assumption 3]), which requires exponentially fast convergence to the stationary measure. Prior works rely on this fast mixing property to handle the non-stationarity of Markovian data when taking conditional expectations, which is ubiquitous in the analysis of SGD-type algorithms. However, our analysis shows that, under the mild Assumption 3.2, a simple change of measures suffices. As uniform geometric ergodicity is typically difficult to verify in real-world scenarios, our results are therefore more broadly applicable. For completeness, in Appendix D.4, we provide an extension of our analysis under the stronger uniform geometric ergodicity assumption.

# 4 Convergence Analysis

## 4.1 Assumptions

To conduct the convergence analysis, we first impose the following assumptions on the objective functions and the stochastic gradients.

**Assumption 4.1.** The global objective function $F$ is $L$-smooth; that is, for all $w_1, w_2 \in \mathbb{R}^d$:

$$F(w_2) \leq F(w_1) + \langle \nabla F(w_1), w_2 - w_1 \rangle + \frac{L}{2} \|w_1 - w_2\|^2.$$

In some cases, we will need the following assumption that requires the sample-wise local objective functions to be $L$-smooth:

**Assumption 4.2.** For every $m \in [M]$, the sample-wise objective functions $f_m(w; x)$ are $L$-smooth; that is, for all $w_1, w_2 \in \mathbb{R}^d, x \in \Omega_m$,

$$f_m(w_2; x) \leq f_m(w_1; x) + \langle \nabla f_m(w_1; x), w_2 - w_1 \rangle + \frac{L}{2} \|w_1 - w_2\|^2.$$

We note that Assumption 4.2 implies Assumption 4.1.

Next, we state an assumption on the noise of the stochastic gradients.

**Assumption 4.3.** For all clients $m \in [M]$, for all $x \in \Omega_m$, and $w \in \mathbb{R}^d$, there exists $\sigma > 0$ such that

$$\|\nabla f_m(w; x) - \nabla F_m(w)\|^2 \leq \sigma^2.$$

Assumption 4.3 uniformly bounds the gradient estimation error for each client. While stronger than the bounded variance assumption, this assumption is standard for stochastic optimization with Markovian noise in both the centralized [3] and federated [37] settings.

Heterogeneity in client objective functions and data distributions is a well-known challenge for FL algorithms based on Local SGD, as it leads to client drift [27]. In order to provide convergence guarantees in the i.i.d. setting, it is necessary to impose constraints on the local gradient norms [60, 64]. Client heterogeneity poses the same difficulty in the Markovian setting. As such, we impose the following bounded gradient dissimilarity (BGD) assumption.

**Assumption 4.4.** There exist $\theta \geq 0$ and $\delta \geq 1$ such that, for all $w \in \mathbb{R}^d$,

$$\frac{1}{M} \sum_{m=1}^{M} \|\nabla F_m(w)\|^2 \leq \theta^2 + \delta^2 \|\nabla F(w)\|^2 \, .$$

This assumption was first introduced in [27] and has since become ubiquitous in the analysis of Local SGD [64]. We highlight that the BGD assumption includes the assumptions in [60, 33]. While weaker heterogeneity assumptions exist in the literature, these are not straightforward to apply for non-convex objectives where the BGD assumption is standard [30]. In Section 4, we establish iteration and communication complexity bounds of Local SGD under these assumptions.

Finally, we define the following three classes of problems for the analysis of Minibatch SGD, Local SGD, and Local SGD with Momentum (SGD-M). Recall from Section 3.2 that the system-level Markov chain is denoted by $X$. We assume throughout that all objective functions are in general non-convex and $F$ is bounded from below by $F^* > -\infty$.

$$\mathcal{F}_1(L, \sigma, \nu_{ps}, C_\infty) := \{(F, X) : \text{Assumptions 3.1, 3.2, 4.1, and 4.3 hold}\}.$$
$$\mathcal{F}_2(L, \sigma, \nu_{ps}, C_\infty) := \{(F, X) : \text{Assumptions 3.1, 3.2, 4.2 and 4.3 hold}\}.$$
$$\mathcal{F}_3(L, \sigma, \theta, \delta, \nu_{ps}, C_\infty) := \{(F, X) : \text{Assumptions 3.1, 3.2, and 4.2 to 4.4 hold}\}.$$

We note that

$$\mathcal{F}_3(L, \sigma, \theta, \delta, \nu_{ps}, C_\infty) \subset \mathcal{F}_2(L, \sigma, \nu_{ps}, C_\infty) \subset \mathcal{F}_1(L, \sigma, \nu_{ps}, C_\infty).$$

In the following sections, we analyze the convergence of Minibatch SGD, Local SGD, and Local SGD-M for the problem classes $\mathcal{F}_1$, $\mathcal{F}_3$, and $\mathcal{F}_2$, respectively. For any $\epsilon > 0$, we derive conditions on the step sizes, the number of local steps $K$, and the number of communication rounds $T$ that ensure convergence to an $\epsilon$-accurate solution, i.e.,

$$\mathbb{E}[\|\nabla F(\hat{w}_T)\|^2] \leq \epsilon^2,$$

where $\hat{w}_T$ is the output of each algorithm, drawn uniformly at random from the iterates $w_0, \ldots, w_{T-1}$.

We use the Landau big-O notation $\mathcal{O}(\cdot)$, $a \vee b$ for $\max\{a, b\}$, $a \wedge b$ for $\min\{a, b\}$, $\Delta_0$ for $F(w_0) - F^*$ and $G_0$ for $\frac{1}{M} \sum_{m=1}^{M} \mathbb{E}\left[\|\nabla F_m(w_0)\|^2\right]$.

## 4.2 Minibatch SGD

We first establish the following upper bound for Minibatch SGD.

**Theorem 4.5.** *For the problem class $\mathcal{F}_1(L, \sigma, \nu_{ps}, C_\infty)$, with global step size $\gamma \leq 1/L$, the iterates of Minibatch SGD satisfy:*

$$\mathbb{E}\left[\|\nabla F(\hat{w}_T)\|^2\right] \leq \mathcal{O}\left(\frac{\Delta_0}{\gamma T} + \frac{C_\infty \sigma^2}{\nu_{ps} M K}\right).$$

The proof of Theorem 4.5 is given in Appendix D.1. The first deterministic term depends on the initial iterate and decays with the same rate as in the i.i.d. setting. The last term arises from the dependence structure and non-stationarity of the Markov data process. In contrast to i.i.d. sampling [60], these terms cannot be controlled by the step size. In particular, Theorem 4.5 reveals that the stochastic gradient noise is amplified by a factor inversely proportional to the spectral gap of the system-level Markov chain.

The following corollary characterizes the communication and sample complexity of Minibatch SGD.

**Corollary 4.6.** *Under the conditions of Theorem 4.5, the required number of local steps and communication rounds for Minibatch SGD to achieve* $\mathbb{E}[\|\nabla F(\hat{w}_T)\|^2] \leq \epsilon^2$ *are:*

$$K = \mathcal{O}\left(\frac{C_\infty \sigma^2}{\nu_{ps} M \epsilon^2}\right), \quad T = \mathcal{O}\left(\frac{L\Delta_0}{\epsilon^2}\right).$$

The proof of Corollary 4.6 is deferred to Appendix D.1.

## 4.3 Local SGD

In the analysis of Local SGD, bounded heterogeneity assumptions are required, even in the i.i.d. setting [27]. We consider the problem class $\mathcal{F}_3(L, \sigma, \theta, \delta, \nu_{ps}, C_\infty)$, which requires sample-wise smoothness (Assumption 4.2) and bounded heterogeneity (Assumption 4.4).

**Theorem 4.7.** *For any* $F \in \mathcal{F}_3(L, \sigma, \theta, \delta, \nu_{ps}, C_\infty)$, *with local step size* $\eta \leq \mathcal{O}\left(\frac{1}{LK\delta^2}\right)$, *the iterates of Local SGD satisfy:*

$$\mathbb{E}\left[\|\nabla F(\hat{w}_T)\|^2\right] \leq \mathcal{O}\left(\frac{\Delta_0}{\eta K T} + \frac{C_\infty \sigma^2}{\nu_{ps} M K} + \frac{LK\eta\left(\theta^2 + \sigma^2\right)}{\delta^2}\right).$$

The proof of Theorem 4.7 is given in Appendix D.2. The convergence behavior of Local SGD closely resembles that of Minibatch SGD, except for the additional third term introduced by the *client drift*. We note that this term can be controlled by having a sufficiently small step size. Similar to Corollary 4.6, the following corollary specifies the communication and sample complexity of Local SGD for both stationary and non-stationary Markov chains.

**Corollary 4.8.** *Under the conditions of Theorem 4.7 and*

$$\eta \leq \mathcal{O}\left(\frac{\delta^2 \epsilon^2}{KL(\theta^2 + \sigma^2)} \wedge \frac{1}{KL\delta^2}\right),$$

*the required number of local steps and communication rounds for Local SGD to achieve* $\mathbb{E}[\|\nabla F(\hat{w}_T)\|^2] \leq \epsilon^2$, *are:*

$$K = \mathcal{O}\left(\frac{C_\infty \sigma^2}{\nu_{ps} M \epsilon^2}\right), \quad T = \mathcal{O}\left(\frac{L\Delta_0}{\epsilon^2}\left(\delta^2 \vee \frac{\theta^2 + \sigma^2}{\delta^2 \epsilon^2}\right)\right).$$

The proof of Corollary 4.8 is also given in Appendix D.2.

## 4.4 Local SGD with Momentum

In the case of i.i.d. sampling, several algorithms have been proposed to mitigate the client drift effect without the need of Assumption 4.4. One line of work focuses on the control variate technique, first introduced by [27]. Recently, [9, 61] have demonstrated that momentum can also be used to mitigate client drift in the i.i.d. setting.

In this section, we study the convergence of a momentum-based variant of Local SGD, named Local SGD-M, detailed in Algorithm 1. The key difference from Local SGD is that the local updates are computed via the convex combination of the gradient estimate for the local objective and the aggregated updates from the previous communication round. This allows us to build a recursive bound for the gradient estimate error, without relying on any heterogeneity assumption to bound the drift caused by local updates as in the case of Local SGD.

We show that, with Markovian data, this technique also mitigates the impact of client drift. In particular, the lower bound on communication complexity in the i.i.d. setting [42] is achieved up to a constant without any heterogeneity assumptions. The convergence result for Local SGD with momentum holds for the broader function class $\mathcal{F}_2(L, \sigma, \nu_{ps}, C_\infty)$, which retains the sample-wise smoothness condition but relaxes the requirement for bounded heterogeneity.

**Theorem 4.9.** *For the problem class* $\mathcal{F}_2(L, \sigma, \nu_{ps}, C_\infty)$, *with the following conditions on the step sizes:*

$$\eta K L \leq \mathcal{O}\left(\frac{1}{\beta} \wedge \sqrt{\frac{C_\infty}{\nu_{ps} M K \beta^2}} \wedge \sqrt{\frac{L\Delta_0}{\beta^3 T G_0}} \wedge \frac{1}{(1-\beta)} \wedge \frac{1}{\beta \gamma L T}\right), \quad \gamma \leq \mathcal{O}\left(\frac{\beta}{L}\right),$$

**Algorithm 1** Local SGD-M

---

**Input:** initial model $w_0$ and gradient estimate $v_0$, local learning rate $\eta$, global learning rate $\gamma$ and momentum $\beta$
**for** $t = 0$ **to** $T - 1$ **do**
    **for** every client $m \in [M]$ in parallel **do**
        Initialize local model $w_t^{(m,0)} = w_t$
        **for** $k = 0$ **to** $K - 1$ **do**
            $v_t^{(m,k)} = \beta \nabla f_m \left( w_t^{(m,k)}; x_t^{(m,k)} \right) + (1 - \beta) v_t$
            $w_t^{(m,k+1)} = w_t^{(m,k)} - \eta v_t^{(m,k)}$
        **end for**
        Communicate $w_t^{m,K}$
    **end for**
    Aggregate: $v_{t+1} = \frac{1}{\eta M K} \sum_{m=1}^{M} \left( w_t - w_t^{(m,K)} \right)$
    Server update: $w_{t+1} = w_t - \gamma v_{t+1}$
**end for**
**Output:** $\hat{w}_T$ sampled uniformly from $w_0, \ldots, w_{T-1}$.

---

*the iterates of Local SGD-M satisfy:*

$$\mathbb{E} \left[ \|\nabla F(\hat{w}_T)\|^2 \right] \leq \mathcal{O} \left( \frac{L \Delta_0}{\beta T} + \frac{C_\infty \sigma^2}{\nu_{ps} M K} \right).$$

The proof of Theorem 4.9 is given in Appendix D.3. The communication and sample complexities of Local SGD-M are characterized in the following corollary.

**Corollary 4.10.** *Under the conditions of Theorem 4.9, the required number of local steps and communication rounds for Local SGD-M to achieve $\mathbb{E}[\|\nabla F(\hat{w}_T)\|^2] \leq \epsilon^2$ are:*

$$K = \mathcal{O} \left( \frac{C_\infty \sigma^2}{\nu_{ps} M \epsilon^2} \right), \quad T = \mathcal{O} \left( \frac{L \Delta_0}{\beta \epsilon^2} \right).$$

The proof of Corollary 4.10 is delegated to Appendix D.3.

### 4.5   Discussion

**Impact of Markovian Data:** Our convergence analysis in Corollaries 4.6, 4.8, and 4.10 reveals the impact of Markovian sampling for smooth non-convex objectives. Due to temporal dependence between samples, the gradient noise is scaled by $C_\infty / \nu_{ps}$, which depends on the spectral properties of the Markov kernel. Moreover, the stochastic gradient estimates computed by the clients are inherently biased. This bias arises from the Markovian structure of the data and, unlike the i.i.d. case, cannot be controlled by having a small or decaying step size. This is due to the fact that the bias contributes to the terms involving the gradient noise $\sigma^2$ in our bounds. Consequently, in order to control these terms and guarantee convergence, the number of local steps $K$ must be sufficiently large.

However, we emphasize that our results remain consistent with the core principles of FL: leveraging more local computation to reduce communication costs. Indeed, a comparison with [42, Table 2] shows that the lower bound established in the i.i.d. setting for communication complexity can be achieved by Minibatch SGD and Local SGD-M. Similar results for the special case of Federated Stochastic Approximation (FSA) can be observed in [37, Corollary 4.5], where additional local computation is required to reduce the third term in [37, Theorem 4.1].

Some prior works in the centralized [3, 15, 14] or the token-passing [17] setting introduce auxiliary hyper-parameters that can be tuned to make the bias vanish, instead of tuning directly the step size. This is achieved:

- in [17, 14] by decomposing the bias into other terms involving delayed gradients, and scaling such delays as $1/\sqrt{T}$. However, this technique requires a constraint on the minimum number of iterations $T$ (analogous to the number of communication rounds in our analysis),

at the order of the mixing time of the Markov chain. Such a constraint is undesirable in the FL setting, where the main goal is to minimize communication. A recent paper on FSA [65] follows the same strategy and indeed needs to impose a lower bound on $T$ proportional to the *maximum mixing time* across all clients.

- in [3, 15] by using minibatch gradient estimates (instead of using just one sample). More precisely, at each iteration, a random batch size is drawn from a truncated geometric distribution with maximum value $m$, requiring a new hyper-parameter. [3, 15] show that the bias scales inversely with $m$, which is then chosen to be scaled with $\sqrt{T}$.

We highlight that the sample complexities obtained in our paper are of the order of $1/(M\epsilon^4)$, matching those achieved in [3, 15, 14, 17], scaled inversely with the number of clients $M$. However, as already mentioned in Section 3.2, our assumptions on the underlying Markov processes are more general. Furthermore, as the terms involving $\sigma^2$ scale with $1/K$, following the approach in [3, 15] and letting $K$ scale with $T$ would indeed drive these terms to zero as $T \to \infty$. Nevertheless, doing so would not lead to any improvement in the overall sample complexity. Further details on this, as well as a more exhaustive discussion on the differences of our work with existing works, are provided in Appendix C.

**Impact of Heterogeneity:** Convergence analysis of Local SGD requires the BGD heterogeneity assumption. The lower bound for communication complexity in the i.i.d. setting can only be achieved within the low-heterogeneity, low-noise regime; i.e., when $\frac{\theta^2+\sigma^2}{\epsilon^2} \leq \delta^4$. In contrast, Local SGD-M successfully mitigates the client drift caused by heterogeneity. In particular, the lower bound on the communication complexity in [42] is achieved without any heterogeneity assumptions, up to a constant of $1/\beta$, while maintaining the same number of samples per communication round $K$. This contrasts with SCAFFLSA [37], which utilizes control variates to mitigate heterogeneity, resulting in lower communication complexity. However, SCAFFLSA needs a higher number of samples per communication round than its vanilla version, FedLSA, and still relies on a heterogeneity assumption around the optimum. Another work on FSA shows the presence of a persistent bias term [59, Theorem 2], which cannot be controlled with increased local computation or even more communication rounds.

**Impact of Number of Clients** $M$**:** The sample complexity of all three algorithms scales with $1/M$, demonstrating the benefits of collaboration in FL with heterogeneous Markovian data streams and smooth non-convex objective functions.

## 5   Numerical Results

In this section, we evaluate the performance of Local SGD, Minibatch SGD, and Local SGD with Momentum on the Beijing Multi-Site Air-Quality dataset [7], which contains hourly measurements from 12 weather stations across China, collected between March 2013 and February 2017. The prediction target is seasonality-adjusted PM2.5 concentration, using other air quality and meteorological indicators as features. In Appendix E, we provide further details for the time series data preprocessing, non-convex objective, and additional experiments to investigate the impact of the Markov chain's spectral gap with synthetic data.

We partition the data temporally, reserving the last 12 months for testing and using the preceding 36 months for training. To simulate a federated setting with a larger number of clients, we create virtual clients by randomly sampling contiguous windows of $n \in \{6, 12\}$ months from the training period. We use a linear regression model with a non-convex regularizer to encourage sparsity, based on preliminary correlation analysis indicating that some features are not informative. The regularization suppresses less relevant features while preserving smoothness for optimization, and is widely used in robust non-convex optimization [57, 20].

**Impact of heterogeneity:** In Figure 1, we plot the trajectories of the gradient norm over the communication rounds for Minibatch SGD, Local SGD, Local SGD-M, and SCAFFOLD [27], which is the first algorithm proposed in the i.i.d. setting to mitigate heterogeneity in FL. We compare these methods under varying numbers of samples, $K$, per communication round. We observe that the performance of Minibatch SGD and Local SGD-M consistently improves as $K$ increases, whereas Local SGD and SCAFFOLD exhibit little to no improvement. This is consistent with our theoretical findings, which identify heterogeneity as a limiting factor as $K$ increases for Local SGD, but not for Minibatch SGD or Local SGD-M. For SCAFFOLD, we argue that its advantage is clearer in the

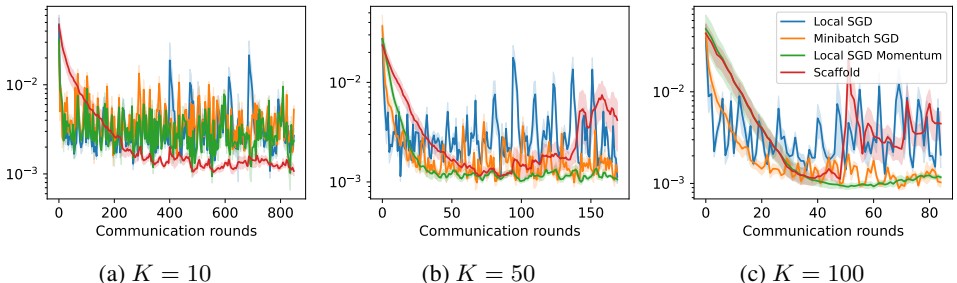

|  |  |  |
|:---:|:---:|:---:|
| (a) $K = 10$ | (b) $K = 50$ | (c) $K = 100$ |

Figure 1: Gradient norm as a function of the number of communication rounds for Local SGD, Minibatch SGD, Local SGD-M, and SCAFFOLD, with $\gamma = 0.1, \eta = 0.01, \beta = 0.5, \lambda = 0.01$ for 120 clients (each client has access to 12 consecutive months of training data) and different numbers of local steps.

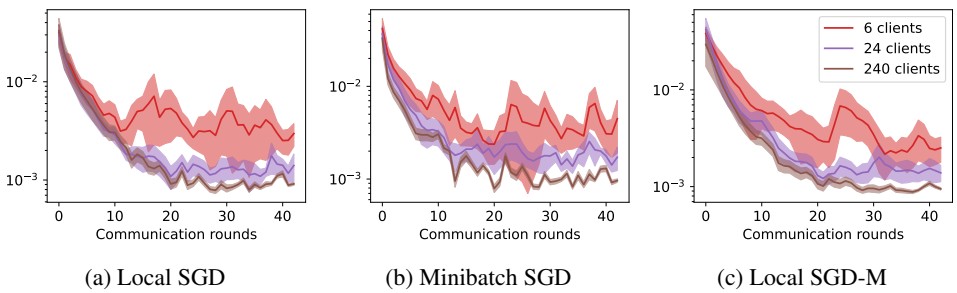

|  |  |  |
|:---:|:---:|:---:|
| (a) Local SGD | (b) Minibatch SGD | (c) Local SGD-M |

Figure 2: Gradient norm as a function of the number of communication rounds for Local SGD, Minibatch SGD & Local SGD-M, with $\gamma = 0.1, \eta = 0.001, \beta = 0.5, \lambda = 0.01$ and $K = 100$ for different numbers of clients. Each client has access to a window of 6 consecutive months of training data.

partial participation setting. [9] also shows that incorporating momentum into SCAFFOLD under the partial client participation setting further improves the convergence. Extending our current analysis to evaluate the combined effect of momentum and control variates in the partial participation setting with Markovian data is a promising direction for future research.

**Impact of Number of Clients $M$:** In Figure 2, we plot the trajectories of the gradient norm for each of the three algorithms with different numbers of clients. For all three methods, the gradient norm decreases as the number of clients increases, indicating that collaboration is beneficial even in the presence of heterogeneous Markovian data—a trend consistent with our theoretical analysis.

## 6 Conclusion

Statistical dependence is a reality in data streams arising from physical and biological systems. A key question is whether collaboration remains beneficial for FL with Markovian data streams. To address this question, we showed via analysis and experiments with pollution data that there is a speed-up in FL with smooth non-convex objectives for Minibatch SGD, Local SGD, and Local SGD with Momentum. However, there is a cost associated with Markovian data streams, quantified by an increase in the sample complexity compared with i.i.d. sampling.

## Acknowledgments and Disclosure of Funding

This research was supported in part by the French government, through the 3IA Côte d'Azur Investments in the Future project managed by the National Research Agency (ANR) with the reference numbers ANR-19-P3IA-0002 and ANR-24-CE25-225, in part by the European Network of Excellence dAIEDGE under Grant Agreement Nr. 101120726, in part by the Groupe La Poste, sponsor of the Inria Foundation, in the framework of the FedMalin Inria Challenge, and in part by the EU HORIZON MSCA 2023 DN project FINALITY (G.A. 101168816). This work was also partially supported by the Inria-Nokia Challenge Learn-Net. Experiments were carried out using the Grid'5000 testbed supported by a scientific interest group hosted by Inria and including CNRS, RENATER and several Universities as well as other organizations.

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

# Appendix

# Contents

# A Preliminaries

## A.1 Pseudo-code for Minibatch SGD and Local SGD

---
**Algorithm 2** Minibatch SGD
---
**Input:** initial model $w_0$, global learning rate $\gamma$.
**for** $t = 0$ **to** $T - 1$ **do**
  **for** every client $m \in [M]$ in parallel **do**
    Receive the global model $w_t$
    **for** $k = 0$ **to** $K - 1$ **do**
      Compute:
      $g_t^{(m,k)} = \nabla f_m\left(w_t; x_t^{(m,k)}\right)$
    **end for**
    Communicate:
    $g_t^m = \frac{1}{K}\sum_{k=0}^{K-1} g_t^{m,k}$
  **end for**
  Aggregate: $g_t = \frac{1}{M}\sum_{m=1}^{M} g_t^m$
  Server update: $w_{t+1} = w_t - \gamma g_t$
**end for**
**Output:** $\hat{w}_T$ sampled uniformly from $w_0, \ldots, w_{T-1}$.

---

---
**Algorithm 3** Local SGD
---
**Input:** initial model $w_0$, local learning rate $\eta$.
**for** $t = 0$ **to** $T - 1$ **do**
  **for** every client $m \in [M]$ in parallel **do**
    Initialize the local model:
    $w_t^{(m,0)} = w_t$
    **for** $k = 0$ **to** $K - 1$ **do**
      $w_t^{(m,k+1)} = w_t^{(m,k)}$
      $\quad - \eta \nabla f_m\left(w_t^{(m,k)}; x_t^{(m,k)}\right)$
    **end for**
    Communicate $w_t^{(m,K)}$
  **end for**
  Server update:
  $w_{t+1} = \frac{1}{M}\sum_{m=1}^{M} w_t^{(m,K)}$
**end for**
**Output:** $\hat{w}_T$ sampled uniformly from $w_0, \ldots, w_{T-1}$.

---

## A.2 Notation

For clarity, we first recall some notations introduced in Section 3.

We use $[M] = \{1, \ldots, M\}$ to denote the set of positive integers up to $M$, and $[K]_0 = [0, K-1]$ to denote the set of non-negative integers less than $K$. For any vector $x \in \mathbb{R}^d$, we write $(x)_i$ for the $i$-th coordinate of $x$.

For every client $m \in [M]$, the data stream $X_m = (x_t^m)_{t \in \mathbb{N}}$ is a time-homogeneous Markov process defined on state space $\Omega_m$ endowed with the corresponding Borel $\sigma$-field $\mathcal{B}_m$ and transition kernel. As an abuse of notation, for all $t \in \mathbb{N}$, and for all $k \in [K]_0$ we denote by $x_t^{(m,k)} := x_{Kt+k}^m$ and by $x_t^m := \left(x_t^{(m,k)}\right)_{k \in [K]_0}$, the $k$-th sample and the set of all $K$ samples used by client $m$ during round $t$, respectively.

The system-level Markov chain $X = (X_m)_{m \in [M]}$ is a product Markov chain defined on $\left(\Omega := \bigtimes_{m \in [M]} \Omega_m, \mathcal{B} := \bigotimes \mathcal{B}_m\right)$, which, at each time step, moves independently on each of $M$ coordinates according to the corresponding transition kernel. Its transition matrix is defined at the Kronecker tensor product $P = \bigotimes_{m \in [M]} P_m$. Similarly, to simplify the notation, we write $x_t^k := \left(x_t^{(m,k)}\right)_{m \in [M]}$ and $x_t := \left(x_t^k\right)_{k \in [K]_0}$.

For a Markov chain, we denote by $\mathbb{P}_q$ and $\mathbb{E}_q$ the corresponding probability distribution and expectation with initial distribution $q$. When $q = \delta(x)$, we simply write $\mathbb{P}_x$ and $\mathbb{E}_x$. We also denote by $\mathcal{F}_t := \sigma\left(x_i^{(m,k)}, m \in [M], k \in [K]_0, i \in [t]_0\right)$ the $\sigma$-algebra generated by all data points received up to round $t$, and write $\mathbb{E}_t[.]$ as an alias for the conditional expectation $\mathbb{E}[.|\mathcal{F}_t]$.

The following notation will be used throughout the Appendix:

$$\bar{g}_t = \frac{1}{MK} \sum_{m,k} \nabla f_m\left(w_t; x_t^{(m,k)}\right),$$

$$g_t = \frac{1}{MK} \sum_{m,k} \nabla f_m\left(w_t^{(m,k)}; x_t^{(m,k)}\right),$$

$$\Delta_t = \mathbb{E}\left[F\left(w_t\right)\right] - F^*,$$

$$s_t = \mathbb{E}\left[\left\|w_{t+1} - w_t\right\|^2\right],$$

where we use $\sum_{m,k}$ for $\sum_{m\in[M]}\sum_{k\in[K]_0}$ unless explicitly stated differently.

With this notation, the global update rule of Minibatch SGD is

$$w_{t+1} = w_t - \gamma\bar{g}_t,$$

the global update rule of Local SGD is

$$w_{t+1} = w_t - \tilde{\eta}_t g_t,$$

where $\tilde{\eta}_t = K\eta_t$, and the global update rule of Local SGD-M is

$$w_{t+1} = w_t - \gamma v_{t+1}.$$

# B Markov Chain Preliminaries

In this section, we introduce some basic properties of Markov chains on general state spaces. In the following, we let $\Omega$ be a Polish space, and $\mathcal{B}$ denote the associated Borel $\sigma$-field.

We begin with the following formal definition of transition kernels.

**Definition B.1.** If $P = \{P(x, A), x \in \Omega, A \in \mathcal{B}\}$ such that:

   (i) for each $A \in \mathcal{B}$, $P(., A)$ is a non-negative measurable function on $\Omega$,

   (ii) for each $x \in \Omega$, $P(x, .)$ is a probability measure on $\mathcal{B}$,

then $P$ is called a transition kernel.

If $\Omega$ is countable, then $P(x, y)$ corresponds to the transition probability of moving from state $x$ to state $y$ in a single step. However, in the case of continuous state spaces, we may have $P(x, y) = 0$ for all $y \in \Omega$, hence the need of defining $P(x, A)$ as the probability of jumping from the current state $x$ into a *subset* $A \subseteq \Omega$.

Now we are ready to define a Markov chain.

**Definition B.2.** A time-homogeneous Markov chain defined on $(\Omega, \mathcal{B})$ with transition kernel $P$ is a sequence of random variables $(X_i)_{i \in \mathbb{N}}$ taking values in $\Omega$ such that for all $i \in \mathbb{N}$:

$$\mathbb{P}(X_i \in A \mid X_{i-1} = x_{i-1}; X_{i-2} = x_{i-2}; \ldots; X_1 = x_1) = \mathbb{P}(X_i \in A \mid X_{i-1} = x_{i-1}) \quad (2)$$
$$= P(x_{i-1}, A).$$

The Markov property (2) can be expressed in the following equivalent form:

**Proposition B.3** (39, Proposition 3.4.3). *Given a time-homogeneous Markov chain $(X_i)_{i \in \mathbb{N}}$ on $(\Omega, \mathcal{B})$, and $f : \Omega \mapsto \mathbb{R}$ a bounded and measurable function, then:*

$$\mathbb{E}[f(X_{n+1}, X_{n+2}, \ldots) \mid X_n = x_n; X_{n-1} = x_{n-1}; \ldots; X_1 = x_1] \quad (3)$$
$$= \mathbb{E}_{x_n}[f(X_1, X_2, \ldots)]$$

We recall the definition of a stationary measure as follows:

**Definition B.4.** The stationary measure of a time-homogeneous Markov chain defined on $(\Omega, \mathcal{B})$ with transition kernel $P$ is a probability measure $\pi$ such that, for all $A \in \mathcal{B}$:

$$\pi(A) = \int_\Omega \mathrm{d}\pi(x) P(x, A),$$

or, equivalently:

$$\mathrm{d}\pi(y) = \int_\Omega \mathrm{d}\pi(x) P(x, \mathrm{d}y).$$

Before formally defining the system-level Markov chain introduced in Section 3.2, we need the following result on the measurability of the product of measurable functions.

**Lemma B.5.** *Let $(\Omega_1, \mathcal{B}_1)$ and $(\Omega_2, \mathcal{B}_2)$ be two measurable spaces. Let $f$ and $g$ be two real-valued measurable functions on $\Omega_1$ and $\Omega_2$, respectively. Let $h$ be a real-valued function defined as:*

$$h : \Omega_1 \times \Omega_2 \to \mathbb{R}$$
$$(x_1, x_2) \mapsto h(x_1, x_2) = f(x_1) g(x_2).$$

*Then $h$ is $\mathcal{B}_1 \otimes \mathcal{B}_2$-measurable.*

*Proof.* Consider $\tilde{f}$ and $\tilde{g}$ defined as:

$$\tilde{f} : \Omega_1 \times \Omega_2 \to \mathbb{R}$$
$$(x_1, x_2) \mapsto \tilde{f}(x_1, x_2) = f(x_1)$$

and

$$\tilde{g} : \Omega_1 \times \Omega_2 \to \mathbb{R}$$
$$(x_1, x_2) \mapsto \tilde{g}(x_1, x_2) = g(x_2).$$

As $f$ and $g$ are $\mathcal{B}_1$-measurable and $\mathcal{B}_2$-measurable, $\tilde{f}$ and $\tilde{g}$ are $\mathcal{B}_1 \otimes \mathcal{B}_2$-measurable. Indeed,

$$\tilde{f}^{-1}\left([a, +\infty)\right) = f^{-1}\left([a, +\infty)\right) \times \Omega_2 \in \mathcal{B}_1 \otimes \mathcal{B}_2, \text{ and}$$
$$\tilde{g}^{-1}\left([a, +\infty)\right) = \Omega_1 \times g^{-1}\left([a, +\infty)\right) \in \mathcal{B}_1 \otimes \mathcal{B}_2.$$

As $h = \tilde{f}\tilde{g}$, and the product of two measurable functions on the same measurable space is measurable [10, Proposition 2.1.7], we proved that $h$ is $\mathcal{B}_1 \otimes \mathcal{B}_2$-measurable. $\square$

The above result can be generalized to the product of more than two functions in a trivial way. We omit the proof for readability.

Now, we provide a formal definition of the system-level Markov chain introduced in Section 3.2. We show that it is indeed a well-defined Markov chain and characterize its transition kernel and stationary distribution.

**Proposition B.6.** *Consider $M$ time-homogeneous Markov chains $X_m, m \in [M]$, each defined on $(\Omega_m, \mathcal{B}_m)$ with transition kernel $P_m$ and stationary distribution $\pi_m$, respectively. The product Markov chain $X$, evolving on $\left(\Omega = \times_{m \in [M]} \Omega_m, \mathcal{B} = \bigotimes_{m \in [M]} \mathcal{B}_m\right)$, with transition kernel $P$ defined as:*

$$P(x, A) = \prod_{m \in [M]} P_m(x_m, A_m),$$

*for all $x = (x_m)_{m \in [M]} \in \Omega$ and $A = \left(\underset{m \in [M]}{\times} A_m\right) \in \mathcal{B}$,*

*is a well-defined Markov chain with stationary distribution $\pi = \bigotimes_{m \in [M]} \pi_m$.*

*Proof.* By construction, it is easy to see that for each $x \in \Omega$, $P(x, .)$ is a product measure on $\mathcal{B}$, hence the second condition in Definition B.4 is satisfied. Also by construction, for every $A \in \mathcal{B}$, $P(., A)$ is non-negative, and Lemma B.5 shows that $P(., A)$ is $\mathcal{B}$-measurable. Thus, $P$ also satisfies the first condition and is therefore a well-defined transition kernel.

We still need to show that the Markov property (2) is achieved. For simplicity, we will do so for the case $M = 2$, but extension to any $M < \infty$ is straightforward.

Denote by $X_1 = \left(X_i^1\right)_{i \in \mathbb{N}}$ and $X_2 = \left(X_i^2\right)_{i \in \mathbb{N}}$ the two chains. Indeed, as the chains are independent, we have that:

$$\mathbb{P}\left[\left(X_i^1, X_i^2\right) \in A_1 \times A_2 \mid \left(X_{i-1}^1, X_{i-1}^2\right) = (x_{i-1}^1, x_{i-1}^2), \dots, \left(X_1^1, X_1^2\right) = (x_1^1, x_1^2)\right]$$
$$= \mathbb{P}\left(X_i^1 \in A_1 \mid X_{i-1}^1 = x_{i-1}^1, \dots, X_1^1 = x_1^1\right) \mathbb{P}\left(X_i^2 \in A_2 \mid X_{i-1}^2 = x_{i-1}^2, \dots, X_1^2 = x_1^2\right)$$
$$= P_1\left(x_{i-1}^1, A_1\right) P_2\left(x_{i-1}^2, A_2\right)$$
$$= P\left((x_{i-1}^1, x_{i-1}^2), (A_1 \times A_2)\right).$$

Since for every $A \in \mathcal{B}$, $P(., A)$ is non-negative and $\mathcal{B}$-measurable, a direct application of Tonelli's Theorem gives us the statement on the stationary measure of the product chain:

$$
\begin{aligned}
\pi(A) &= \left( \bigotimes_{m \in [M]} \pi_m \right)(A) = \prod_{m \in [M]} \pi_m (A_m) \\
&= \int_{\Omega_1} \mathrm{d}\pi_1(x_1) P_1(x_1, A_1) \cdots \int_{\Omega_M} \mathrm{d}\pi_M(x_M) P_M(x_M, A_M) \\
&= \int_{\Omega_1} \cdots \int_{\Omega_m} P_1(x_1, A_1) \ldots P_M(x_M, A_M)\, \mathrm{d}\pi_1(x_1) \ldots \mathrm{d}\pi_M(x_M) \\
&= \int_{\Omega} \mathrm{d}\pi(x)\, P(x, A).
\end{aligned}
$$

$\square$

We recall the definition of total variation distance:

**Definition B.7.** For two probabilities $\mu, \nu$ defined on $(\Omega, \mathcal{B})$, the total variation distance between $\mu$ and $\nu$ is defined as:

$$
\|\mu - \nu\|_{TV} = \sup_{A \in \mathcal{B}} |\mu(A) - \nu(A)| = \frac{1}{2} \int_{\Omega} |\mathrm{d}\mu - \mathrm{d}\nu|.
$$

Below, we give an equivalent definition of the total variation distance:

**Proposition B.8.** *Let $(Y, Z)$ be a coupling of $\mu$ and $\nu$, i.e. the marginal distribution of $Y$ is $\mu$ and the marginal distribution of $Z$ is $\nu$. Then:*

$$
\|\mu - \nu\|_{TV} = \inf \{ \mathbb{P}(Y \neq Z), (Y, Z) \text{ is a coupling of } \mu, \nu \}. \tag{4}
$$

*The coupling $(Y, Z)$ that satisfies $\|\mu - \nu\|_{TV} = \mathbb{P}(Y \neq Z)$ is then called the maximal coupling of $\mu$ and $\nu$.*

The proof can be found in [48, Proposition 3].

Now we present some nice properties on the total variation distance with respect to the stationary measure of Markov chains:

**Proposition B.9.** *Let $X_1, X_2, \ldots$ be a Markov chain on $(\Omega, \mathcal{B})$ with transition kernel $P$ and stationary distribution $\pi$, then:*

  *(i) For all probability distributions $\mu, \nu$ defined over $(\Omega, \mathcal{B})$:*

$$
\|\mu P - \nu P\|_{TV} \leq \|\mu - \nu\|_{TV}.
$$

  *(ii) More specifically, advancing the chain will not increase the total variation distance to the stationary distribution, i.e.:*

$$
\|P^n(x, .) - \pi(.)\|_{TV} \leq \|P^{n-1}(x, .) - \pi(.)\|_{TV}. \tag{5}
$$

  *(iii) Let $d(n) = 2 \sup_{x \in \Omega} \|P^n(x, .) - \pi(.)\|_{TV}$. Then for $\mu$ varying over probability distribution defined over $(\Omega, \mathcal{B})$:*

$$
d(n) = 2 \sup_{\mu} \|\mu P^n - \pi\|_{TV}. \tag{6}
$$

  *and $d(.)$ is sub-multiplicative, i.e.:*

$$
d(m + n) \leq d(m) d(n) \quad \text{for } m, n \in \mathbb{N}. \tag{7}
$$

  *(iv) For $i \in [n]$, let $\mu_i$ and $\nu_i$ be probability distributions on $(\Omega_i, \mathcal{B}_i)$. Define $\mu := \bigotimes_{i=1}^{n} \mu_i$ and $\nu := \bigotimes_{i=1}^{n} \nu_i$ as two probability distributions on $(\Omega := \bigtimes_{i=1}^{n} \Omega_i, \mathcal{B} := \bigotimes_{i=1}^{n} \mathcal{B}_n)$. Then we have:*

$$
\|\mu - \nu\|_{TV} \leq \sum_{i=1}^{n} \|\mu_i - \nu_i\|_{TV}.
$$

*Proof.* The proof of $(i), (ii)$ and $(iii)$ can also be found in [48, Proposition 3]. Here we present the proof for the last statements.

By using (4), it follows that

$$\|\mu_m - \nu_m\|_{TV} = \mathbb{P}_m\{X_m \neq Y_m\},$$

where, for all $m \in [M]$, $(X_m, Y_m)$ is the maximal coupling of $\mu_m, \nu_m$. Define $X = \{X_m\}_{m\in[M]}$ and $Y = \{Y_m\}_{m\in[M]}$. Then again by (4) we have

$$\begin{aligned}
\|\mu - \nu\|_{TV} &\leq \mathbb{P}\{X \neq Y\} \\
&= \mathbb{P}\left\{\bigcup_{m=1}^{M} X_m \neq Y_m\right\} \\
&\leq \sum_{m=1}^{M} \mathbb{P}_m\{X_m \neq Y_m\} = \sum_{m=1}^{M} \|\mu_m - \nu_m\|_{TV},
\end{aligned}$$

which follows from the union bound. $\qquad\square$

We now give the definition of uniform geometric ergodicity, which is commonly used in the literature of stochastic optimization with Markovian data [3, 37].

**Definition B.10.** A Markov chain defined on $(\Omega, \mathcal{B})$ with transition kernel $P$ is uniformly geometrically ergodic if:

$$\sup_{x\in\Omega} \|P^n(x,.) - \pi\|_{TV} \leq c\rho^n,$$

for some $c \leq \infty$ and $\rho < 1$.

Uniform geometric ergodicity implies that, regardless of the starting distribution, the chain converges exponentially fast to its stationary measure in terms of total variation norm.

To end this section, we have the following results on the mixing time of Markov chains.

**Proposition B.11.** *Given a Markov chain defined on $(\Omega, \mathcal{B})$, let $d(t)$ the total variation distance defined in* (6)*, the mixing time $\tau(\epsilon)$ and $\tau$ are defined as:*

$$\tau(\epsilon) = \min\{t : d(t) \leq \epsilon\}, \tag{8}$$
$$\tau = \tau(1/4),$$

*For any positive integer $n$, we have*
$$d(n\tau(\epsilon)) \leq \epsilon^n.$$

*More specifically,*
$$d(n\tau) \leq 4^{-n},$$

*and*
$$\tau(\epsilon) \leq \lceil \log_4(\epsilon^{-1}) \rceil \tau.$$

*Proof.* The first statement is a direct application of (7). Simply by taking $\epsilon = 1/4$, we obtain the second statement. Then with $n \geq \lceil \log_4(\epsilon^{-1}) \rceil$, $d(n\tau) \leq \epsilon$, which implies that $n\tau \geq \tau(\epsilon)$, hence the last statement. $\qquad\square$

**Proposition B.12.** *Consider the product Markov chain in Proposition B.6. The mixing time of this product Markov chain $\tau$ (see definitions in Proposition B.11) satisfies*

$$\tau \leq (\lceil \log_4 M \rceil + 1) \max_{m\in[M]} \tau_m.$$

*Proof.* Let

$$d(t) = 2\sup_{x \in \Omega} \left\| P^t\left(x, .\right) - \pi \right\|,$$

$$d_m(t) = 2\sup_{x \in \Omega_m} \left\| P_m^t\left(x, .\right) - \pi_m \right\|.$$

By Proposition B.9, we have

$$d(t) \leq \sum_{m=1}^M d_m(t). \tag{9}$$

As in (5), advancing the Markov chain in time can only move it closer to the stationary distribution. Thus, for all $m \in [M]$,

$$d_m\left(\max_{m \in [M]} \tau_m\left(\frac{\epsilon}{M}\right)\right) \leq \frac{\epsilon}{M},$$

where $\tau\left(\epsilon\right) := \min\left\{t : d(t) \leq \epsilon\right\}$ and $\tau_m\left(\epsilon\right) = \min\left\{t : d_m(t) \leq \epsilon\right\}$. Hence, from (9),

$$\tau\left(\epsilon\right) \leq \max_{m \in [M]} \tau_m\left(\frac{\epsilon}{M}\right).$$

Then from Proposition B.11, we have that, for all $m \in [M]$,

$$\tau_m\left(\epsilon/M\right) \leq \lceil \log_4\left(M/\epsilon\right) \rceil \tau_m.$$

The lemma follows by taking $\epsilon = 1/4$ as in the definition of mixing time (8). $\qquad \square$

## C  Comparison with Existing Work

In Table 1, we compare our results with existing ones in stochastic non-convex optimization with Markovian data in the centralized (or token-passing) setting [3, 15, 14, 17] setting. In the FL setting with i.i.d. data, we include the lower bounds in [42], along with some results in [9]. For completeness, we also report results in the literature of Federated Stochastic Approximation (FSA), which is analogous to stochastic optimization with strongly convex [65] or linear least-square [37] objectives.

Table 1: Comparison of algorithms in terms of communication and sample complexity to reach an $\epsilon$-stationary point for smooth, non-convex objectives.
**Additional Constraints:** GE: uniform geometric ergodicity (Definition B.10), ST: stationary Markov processes, BG: bounded gradient, BD: bounded domain, FS: finite state space, LT: lower bound on $T$, SS: sample-wise smoothness (Assumption 4.2), BR: bounded Radon–Nikodym derivative (Assumption 3.2), BH: bounded heterogeneity (Assumption 4.4), BHO: bounded heterogeneity around optimum (i.e., the average distance between the optimum of the local objectives and the global objective is bounded, see 37, Page 4), LS: linear least-square objectives, SC: strongly convex global objective..

| | Algorithm | Communication $T$ | Sample complexity $KT$ | Additional Constraints |
|---|---|---|---|---|
| **Centralized[1]** | [3, Thm. 3] | | $\tilde{\mathcal{O}}\left(\tau\left[\frac{L\Delta_0}{\epsilon^2} + \frac{L\Delta_0\sigma^2}{\epsilon^4}\right]\right)$ | GE |
| | MAG [15, Thm. 4.3] | | $\tilde{\mathcal{O}}\left(\tau\frac{(M+L+G)^2 G^2}{\epsilon^4}\right)$ (2) | BG, BD |
| | MC-SGD [17, Thm. 5.3] | | $\tilde{\mathcal{O}}\left(\tau\left[\frac{L\Delta_0+\sigma^2}{\epsilon^2} + \frac{(L\Delta_0+\sigma^2)\sigma^2}{\epsilon^4}\right]\right)$ | FS, GE, LT |
| | [14, Thm. 3] | | $\tilde{\mathcal{O}}\left(\tau\frac{L^2(1+\|w^*\|^2+\|w_0-w^*\|^2)}{\epsilon^4}\right)$ (3) | FS, GE, LT |
| **FL i.i.d.** | Lower bound [42] | $\Omega\left(\frac{L\Delta_0}{\epsilon^2}\right)$ | $\Omega\left(\frac{L\Delta_0\sigma}{M\epsilon^3}\right)$ (4) | |
| | FedAvg-M [9, Thm. 1] | $\mathcal{O}\left(\frac{L\Delta_0}{\epsilon^2}\right)$ | $\mathcal{O}\left(\frac{L\Delta_0\sigma^2}{M\epsilon^4}\right)$ | |
| | FedAvg-M-VR [9, Thm. 2] | $\mathcal{O}\left(\frac{L\Delta_0}{\epsilon^2}\right)$ | $\mathcal{O}\left(\frac{L\Delta_0\sigma}{M\epsilon^3}\right)$ (5) | SS |
| **FSA[6]** | FedLSA [37, Cor. 4.5] | $\mathcal{O}\left(\frac{1}{\epsilon}\log\frac{1}{\epsilon}\right)$ | $\tilde{\mathcal{O}}\left(\frac{\tau_{\max}}{M\epsilon^2}\log\frac{1}{\epsilon}\right)$ | GE, ST, BHO, LS |
| | SCAFFLSA [37, Cor. 5.2] | $\mathcal{O}\left(\log\frac{1}{\epsilon}\right)$ | $\mathcal{O}\left(\frac{1}{M\epsilon^2}\log\frac{1}{\epsilon}\right)$ (7) | BHO, LS |
| | FedHSA [65, Cor. 1] | $\tilde{\mathcal{O}}\left(\frac{1}{\epsilon}\right)$ | $\tilde{\mathcal{O}}\left(\frac{\tau_{\max}}{M\epsilon^2}\right)$ | FS, GE, SC, LT |
| **Ours** | **Minibatch SGD** | $\mathcal{O}\left(\frac{L\Delta_0}{\epsilon^2}\right)$ | $\mathcal{O}\left(\frac{L\Delta_0 C_\infty\sigma^2}{\nu_{ps}M\epsilon^4}\right)$ | BR |
| | **Local SGD** | $\mathcal{O}\left(\frac{L\Delta_0}{\epsilon^2}\left(\delta^2 \vee \frac{\theta^2+\sigma^2}{\delta^2\epsilon^2}\right)\right)$ | $\mathcal{O}\left(\frac{L\Delta_0 C_\infty\sigma^2}{\nu_{ps}M\epsilon^4}\left(\delta^2 \vee \frac{\theta^2+\sigma^2}{\delta^2\epsilon^2}\right)\right)$ | BR, BH, SS |
| | **Local SGD-M** | $\mathcal{O}\left(\frac{L\Delta_0}{\beta\epsilon^2}\right)$ | $\mathcal{O}\left(\frac{L\Delta_0 C_\infty\sigma^2}{\beta\nu_{ps}M\epsilon^4}\right)$ | BR, SS |

$\tilde{\mathcal{O}}$ ignores logarithmic constants.

[1] In the centralized (or token-passing) setting, $\tau$ is the mixing time of the underlying Markov process.

[2] $M$ and $G$ are upper bounds on the domain diameter and on the gradient norm, respectively.

[3] $w^*$ is a global optimum of the objective. Their proof also requires a projection onto the unknown set that contains all the optima, and hides exponential factors in $\tau$.

[4] The setting for the lower bound requires other assumptions. We refer to the original paper [42] for a complete list.

[5] The variance-reduction technique is akin to the STORM algorithm [11], which relies on the sample-wise smoothness assumption (Assumption 4.2) and achieves the sample complexity lower bound in [42]. Extending Local SGD-M with variance reduction to attain this bound is a promising direction.

[6] This setting focuses on stochastic approximation problems, with strongly monotone or linear operators. The rates presented here denote the complexity to reach an $\epsilon$-*optimal solution*. We omit problem-specific constants, except $\tau_{\max}$ (the maximum mixing time across all clients) and $M$.

[7] The authors present only the rate for the i.i.d. case.

We first give a theoretical justification that clarifies why adjusting the step size alone offers limited leverage for controlling the gradient noise term. By a standard decomposition from the $L$-smoothness condition:

$$\mathbb{E}_t[F(w_{t+1})] \leq F(w_t) - \gamma/2(1 - 2\gamma L)||\nabla F(w_t)||^2 + \gamma/2||E_t[g_t] - \nabla F(w_t)||^2$$
$$+ \gamma^2 L E_t[||g_t - \nabla F(w_t)||^2]$$

Let $b_t = ||\mathbb{E}_t[g_t] - \nabla F(w_t)||^2$ and $v_t = \mathbb{E}_t\left[||g_t - \nabla F(w_t)||^2\right]$ be the bias and the variance of the gradient estimate. This bias arises from the Markovian structure of the data, since in the i.i.d. setting, $\mathbb{E}_t[g_t] = \nabla F(w_t)$ for all $t$. Observe that the bias is scaled by the step size $\gamma$. Since the term $||\nabla F(w_t)||^2$ is also scaled by $\gamma$, the bias cannot be directly controlled by the choice of step size without sacrificing the reduction of the objective. Note that by Jensen's inequality, $b_t \leq v_t$, hence the bias also contributes to the "variance" term arising in our main results. This leads to a non-vanishing term involving the gradient noise $\sigma^2$ in our results as the step size decreases.

We now give additional details of the techniques used in prior works to control this bias, and obtain a vanishing "variance" term. In short, they introduce other hyper-parameters that can be tuned to make the bias vanish, instead of tuning directly the step size directly.

First, [17, 14, 65] decompose the bias term into other terms containing delayed gradient. In detail, in order to bound the bias, instead of taking the conditional expectation up to the current iteration $t$, they condition on a *distant past iteration*, which is $t$ minus some delay. This delay is later chosen to be scaled with $1/\sqrt{T}$ [17], or with the mixing time $\tau(\epsilon)$ (see Proposition B.11), where $\epsilon$ is then chosen to be scale with the step size [14, 65] (the step size itself is tuned to be at the order of $1/T$). This technique requires a constraint on the minimum number of iterations (or communication rounds) $T$ at the order of $\tilde{\mathcal{O}}(\tau_{\mathrm{mix}})$ (this constraint is referred to as LT in Table 1). As revealed in Table 1, [14] also contains hidden factors that can be exponential in $\tau_{\mathrm{mix}}$. The bound for non-convex functions in [14] (Theorem 3 in the arxiv version) also requires a projection onto the unknown set that contains all the optima.

Next, [3, 15] use minibatch gradient estimates. Precisely, at each iteration, a random batch size is drawn from a truncated geometric distribution. Hence, a new hyper-parameter is introduced, which is the truncation threshold $m$. The authors of [3, 15] show that $b_t$ scales inversely with $m$, which is then chosen to be scaled with $\sqrt{T}$.

In the FSA setting, the analysis in [37] employs a blocking technique that sub-samples data sequences to mitigate temporal correlations, using a sub-sampling gap at the order of the maximum mixing time $\tau_{\mathrm{max}}$. Beyond assuming uniform geometric ergodicity, this approach also requires the underlying Markov processes to be stationary. Moreover, their analysis is restricted to linear least-squares problems.

We emphasize that the gradient noise term in our results can be easily made to vanish as $T$ diverges. Consider Minibatch SGD, under the setting of Theorem 4.5, By choosing $K = \mathcal{O}\left(\frac{C_\infty T}{\nu_{ps} M}\right)$, then we have:
$$\mathbb{E}\left[||\nabla F(\hat{w}_T)||^2\right] \leq \mathcal{O}\left(\frac{L\Delta_0}{T} + \frac{\sigma^2}{T}\right),$$
which leads to the sample complexity:
$$KT = \mathcal{O}\left(\frac{C_\infty(L\Delta_0 \vee \sigma^2)^2}{\nu_{ps} M \epsilon^4}\right).$$

However, the above result shows no improvement in sample complexity compared to those reported in the main paper. We highlight that our complexity results match the best-known rates achieved in the centralized setting [3, 15, 17, 14], scaled by the number of clients $M$ (see Table 1), while avoiding the commonly imposed uniform geometric ergodicity assumption. Instead, we rely on a general mild assumption about the boundedness of the Radon-Nikodym derivative (Assumption 3.2), which allows for a simple change of measures to handle the non-stationarity of Markovian samples when taking conditional expectation. Prior works rely heavily on the exponential mixing property to deal with non-stationarity. The analysis in [3] even yields a dependence on the square of the mixing time [3, Section B.1]. The proof in [15] does not require uniform geometric mixing, but in the other hand relies strongly on the boundedness of the domain. We decide to choose $K$ as independent of both the step size and the number of rounds $T$, because in real-world scenarios, the value of $K$ can be dictated by other factors such as the data-arrival rate and the client's buffer capacity.

## D   Convergence Analysis

We begin this section with the following lemma that bounds the square error of the stochastic gradient estimate computed by $M$ clients, each using $K$ Markovian samples.

**Lemma D.1.** *Let $q$ be the initial measure of the system-level Markov chain defined in Section 3.2. If Assumptions 3.1, 3.2 and 4.3 hold, then we have:*

$$\mathbb{E}_t\left[\left\|\frac{1}{MK}\sum_{m,k}\nabla f_m\left(w_t;x_t^{(m,k)}\right)-\nabla F(w_t)\right\|^2\right] \leq \frac{4C_\infty\sigma^2}{\nu_{ps}MK}.$$

*Proof.* By Markov's property (3):

$$\mathbb{E}_t\left[\left\|\frac{1}{MK}\sum_{m,k}\nabla f_m\left(w_t;x_t^{(m,k)}\right)-\nabla F(w_t)\right\|^2\right]$$

$$= \mathbb{E}_{x_{t-1}^{K-1}}\left[\left\|\frac{1}{MK}\nabla f_m\left(w_t;x_t^{(m,k)}\right)-\nabla F(w_t)\right\|^2\right].$$

Hence, by change of measures:

$$\mathbb{E}_t\left[\left\|\frac{1}{MK}\sum_{m,k}\nabla f_m\left(w_t;x_t^{(m,k)}\right)-\nabla F(w_t)\right\|^2\right]$$

$$= \mathbb{E}_\pi\left[\frac{dP\left(x_{t-1}^{K-1},.\right)}{d\pi}\left\|\frac{1}{MK}\sum_{m,k}\nabla f_m\left(w_t;x_t^{(m,k)}\right)-\nabla F(w_t)\right\|^2\right]$$

$$\leq \left\|\frac{dP(x_{t-1}^{K-1},.)}{d\pi}\right\|_{\pi,\infty}\mathbb{E}_\pi\left[\left\|\frac{1}{MK}\sum_{m,k}\nabla f_m\left(w_t;x_t^{(m,k)}\right)-\nabla F(w_t)\right\|^2\right] \quad \text{(Holder's inequality)}$$

$$\leq C_\infty\mathbb{E}_\pi\left[\left\|\frac{1}{MK}\sum_{m,k}\nabla f_m\left(w_t;x_t^{(m,k)}\right)-\nabla F(w_t)\right\|^2\right] \quad \text{(Assumption 3.2).} \tag{10}$$

We now utilize [43, Theorem 3.7], applied on the stationary system-level Markov process:

$$\mathbb{E}_\pi\left[\left\|\frac{1}{MK}\sum_{m,k}\nabla f_m\left(w_t;x_t^{(m,k)}\right)-\nabla F(w_t)\right\|^2\right]$$

$$= \sum_{j=1}^d\mathbb{E}_\pi\left[\left(\frac{1}{MK}\sum_{m,k}\nabla f_m\left(w_t;x_t^{(m,k)}\right)-\nabla F(w_t)\right)_j^2\right]$$

$$\overset{(i)}{\leq} \frac{4}{K\nu_{ps}}\sum_{j=1}^d\mathbb{E}_\pi\left[\left(\frac{1}{M}\sum_m\nabla f_m\left(w_t;x\right)-\nabla F(w_t)\right)_j^2\right] \tag{11}$$

$$\leq \frac{4}{K\nu_{ps}}\mathbb{E}_\pi\left[\left\|\frac{1}{M}\sum_m\nabla f_m(w_t;x)-\nabla F(w_t)\right\|^2\right]$$

$$\overset{(ii)}{\leq} \frac{4\sigma^2}{MK\nu_{ps}}.$$

In $(i)$, we apply [43, Theorem 3.7] to each of $d$ functions:

$$h_i : \Omega \to \mathbb{R},$$

$$h_i(x_t^k) = \left( \frac{1}{M} \sum_m \nabla f_m(w_t; x_t^{(m,k)}) \right)_i.$$

In $(ii)$, we use the fact that $\mathbb{E}_{\pi_m}[\nabla f_m(w; x)] = \nabla F_m(w)$ together with the independence between clients' Markov processes. As such, the term $\mathbb{E}_\pi\left[\|1/M \sum_m \nabla f_m(w; x) - \nabla F(w)\|^2\right]$ can be seen as *the variance of the sum of $M$ independent random variables*. Using Assumption 4.3, it follows that:

$$\mathbb{E}_\pi\left[ \left\| \frac{1}{M} \sum_m \nabla f_m(w; x) - \nabla F(w) \right\|^2 \right] = \frac{1}{M^2} \sum_m \mathbb{E}_{\pi_m}\left[ \|\nabla f_m(w; x) - \nabla F_m(w)\|^2 \right]$$

$$\leq M\sigma^2$$

$\square$

Hence, by plugging (11) back into (10), we prove the lemma.

### D.1 Minibatch SGD

In this section, we consider the problem class $\mathcal{F}_1(L, \sigma, \tau)$. We begin with the following descent lemma.

**Lemma D.2.** *For the problem class $\mathcal{F}_1(L, \sigma, \tau)$, with constant global step size $\gamma \leq \frac{1}{L}$, the iterates of Minibatch SGD satisfy:*

$$\Delta_{t+1} \leq \Delta_t - \frac{\gamma}{2}\mathbb{E}\left[\|\nabla F(w_t)\|^2\right] + \frac{2\gamma C_\infty \sigma^2}{\nu_{ps} M K}$$

*Proof.* Since $F$ is $L$-smooth,

$$F(w_{t+1}) \leq F(w_t) + \langle \nabla F(w_t), w_{t+1} - w_t \rangle + \frac{L}{2}\|w_{t+1} - w_t\|^2$$

$$= F(w_t) - \gamma\langle \nabla F(w_t), \bar{g}_t \rangle + \frac{L}{2}\|w_{t+1} - w_t\|^2.$$

(12)

Note that the term $-\gamma\langle \nabla F(w_t), \bar{g}_t \rangle$ can be expanded as

$$-\gamma\langle \nabla F(w_t), \bar{g}_t \rangle = -\frac{\gamma}{2}\|\nabla F(w_t)\|^2 + \frac{\gamma}{2}\|\bar{g}_t - \nabla F(w_t)\|^2 - \frac{\gamma}{2}\|\bar{g}_t\|^2$$

(13)

$$= -\frac{\gamma}{2}\|\nabla F(w_t)\|^2 + \frac{\gamma}{2}\|\bar{g}_t - \nabla F(w_t)\|^2 - \frac{1}{2\gamma}\|w_{t+1} - w_t\|^2.$$

Plugging (13) back into (12), subtracting $F^*$ from both sides, and taking the conditional expectation, we then have

$$\mathbb{E}_t[F(w_{t+1} - F^*] \leq \mathbb{E}_t[F(w_t) - F^*] - \frac{\gamma}{2}\mathbb{E}_t\left[\|\nabla F(w_t)\|^2\right] + \frac{\gamma}{2}\mathbb{E}_t\left[\|\bar{g}_t - \nabla F(w_t)\|^2\right]$$ (14)

$$- \left(\frac{1}{2\gamma} - \frac{L}{2}\right)\mathbb{E}_t\left[\|w_{t+1} - w_t\|^2\right].$$

Then, with the condition $\gamma \leq \frac{1}{L}$, the last term in (14) can be ignored. The lemma follows by replacing the second term in (14) using Lemma D.1, then taking the full expectation. $\square$

Now we proceed to the proof of Theorem 4.5.

**Theorem D.3** (Theorem 4.5 in the main paper). *For the problem class $\mathcal{F}_1(L, \sigma, \tau)$, with global step size $\gamma \leq 1/L$, the iterates of Minibatch SGD satisfy:*

$$\mathbb{E}\left[\|\nabla F(\hat{w}_t)\|^2\right] \leq \mathcal{O}\left(\frac{\Delta_0}{\gamma T} + \frac{C_\infty \sigma^2}{MK}\right)$$

*Proof.* The proof of Theorem D.3 follows directly from Lemma D.2 by rearranging the terms, then taking the average from $t = 0$ to $T - 1$

$$\frac{1}{T}\sum_{t=0}^{T-1}\mathbb{E}\left[\|\nabla F(w_t)\|^2\right] \leq \frac{2\Delta_0}{\gamma T} + \frac{4C_\infty \sigma^2}{\nu_{ps}MK} \tag{15}$$

Noticing that with $\hat{w}_T$ sampled uniformly from $w_0, \ldots w_{T-1}$,

$$\frac{1}{T}\sum_{t=0}^{T-1}\mathbb{E}\left[\|\nabla F(w_t)\|^2\right] = \mathbb{E}\left[\|\nabla F(\hat{w}_T)\|^2\right],$$

which concludes the proof. $\qquad\square$

Now we are ready to prove Corollary 4.6.

**Corollary D.4** (Corollary 4.6 in the main paper). *Under the conditions of Theorem D.3, to achieve $\mathbb{E}[\|\nabla F(\hat{w}_T)\|^2] \leq \epsilon^2$, the required number of local steps and communication rounds of Minibatch SGD are:*

$$K = \mathcal{O}\left(\frac{C_\infty \sigma^2}{\nu_{ps}M\epsilon^2}\right), \quad T = \mathcal{O}\left(\frac{L\Delta_0}{\epsilon^2}\right)$$

*Proof.* By choosing

$$K \geq \frac{8C_\infty \sigma^2}{\nu_{ps}M\epsilon^2},$$

$$T \geq \frac{4\Delta_0}{\gamma\epsilon^2} \geq \frac{4L\Delta_0}{\epsilon^2},$$

(15) guarantees that $\mathbb{E}\left[\|\nabla F(\hat{w}_T)\|^2\right] \leq \epsilon^2$. $\qquad\square$

### D.2 Local SGD

For the analysis of Local SGD, we define

$$\xi_t = \frac{1}{MK}\sum_{m,k}\mathbb{E}_t\left[\left\|w_t^{(m,k)} - w_t\right\|^2\right],$$

where $\xi_t$ is the client drift observed in the local computation phase of the $t$-th communication round.

We recall that the analysis of Local SGD is for the problem class $\mathcal{F}_3(L, \sigma, \theta, \delta, \tau)$. We begin with the following descent lemma.

**Lemma D.5.** *If the local step size satisfies*

$$\tilde{\eta} \leq \frac{1}{10L\delta^2} < \frac{1}{L},$$

*then the iterates of Local SGD satisfy*

$$\Delta_{t+1} \leq \Delta_t - \frac{45\tilde{\eta}}{98}\mathbb{E}\left[\|\nabla F(w_t)\|^2\right] + \frac{4\tilde{\eta}C_\infty \sigma^2}{\nu_{ps}MK} + \frac{40}{98}\frac{L\tilde{\eta}^2\left(\theta^2 + \sigma^2\right)}{\delta^2}.$$

*Proof.* We can adapt the proof of Lemma D.2 to Local SGD but with the update rule $w_{t+1} = w_t - \tilde{\eta} g_t$:

$$\mathbb{E}_t[F(w_{t+1}) - F^*] \leq \mathbb{E}_t[F(w_t) - F^*] - \frac{\tilde{\eta}}{2}\mathbb{E}_t\left[\|\nabla F(w_t)\|^2\right] + \frac{\tilde{\eta}}{2}\mathbb{E}_t\left[\|g_t - \nabla F(w_t)\|^2\right]$$
$$- \left(\frac{1}{2\tilde{\eta}} - \frac{L}{2}\right)\mathbb{E}_t\left[\|w_{t+1} - w_t\|^2\right]$$
$$\leq \mathbb{E}_t[F(w_t) - F^*] - \frac{\tilde{\eta}}{2}\mathbb{E}_t\left[\|\nabla F(w_t)\|^2\right] + \tilde{\eta}\mathbb{E}_t\left[\|\bar{g}_t - \nabla F(w_t)\|^2\right]$$
$$+ \tilde{\eta}\mathbb{E}_t\left[\|g_t - \bar{g}_t\|^2\right]. \tag{16}$$

Here in the last step, we use the triangle inequality, and the condition on the step size $\tilde{\eta} \leq \frac{1}{10L\delta^2} < \frac{1}{L}$.
Using the sample-wise smoothness (Assumption 4.2) yields:

$$\mathbb{E}_t\left[\|g_t - \bar{g}_t\|^2\right] = \mathbb{E}_t\left[\left\|\frac{1}{MK}\sum_{m,k}\nabla f_m\left(w_t^{(m,k)}; x_t^{(m,k)}\right) - \nabla f_m\left(w_t; x_t^{(m,k)}\right)\right\|^2\right]$$
$$\leq \frac{L^2}{MK}\sum_{m,k}\mathbb{E}_t\left[\left\|w_t^{(m,k)} - w_t\right\|^2\right] \tag{17}$$
$$= L^2\xi_t.$$

where Jensen's inequality is used in the last step.

The local drift $\xi_t$ can then be bounded as follows:

$$\xi_t = \frac{1}{MK}\sum_{m,k}\mathbb{E}_t\left[\left\|w_t^{(m,k)} - w_t\right\|^2\right]$$
$$= \frac{1}{MK}\sum_{m,k}\mathbb{E}_t\left[\left\|\sum_{j=0}^{k-1}\eta\nabla f_m\left(w_t^{(m,j)}; x_t^{(m,j)}\right)\right\|^2\right]$$
$$\leq \frac{1}{MK}\sum_{m,k}\eta^2 k\sum_{j=0}^{k-1}\mathbb{E}_t\left[\left\|\nabla f_m\left(w_t^{(m,j)}; x_t^{(m,j)}\right)\right\|^2\right]$$
$$\leq \frac{\tilde{\eta}^2}{MK^2}\sum_{m,k}\sum_{j=0}^{k-1}\mathbb{E}_t\left[\left\|\nabla f_m\left(w_t^{(m,j)}; x_t^{(m,j)}\right)\right\|^2\right]$$
$$\leq \frac{\tilde{\eta}^2}{MK^2}\sum_{m,k}\sum_{j=0}^{K-1}\mathbb{E}_t\left[\left\|\nabla f_m\left(w_t^{(m,j)}; x_t^{(m,j)}\right)\right\|^2\right]$$
$$\leq \frac{\tilde{\eta}^2}{MK}\sum_{m,j}\mathbb{E}_t\left[\left\|\nabla f_m\left(w_t^{(m,j)}; x_t^{(m,j)}\right)\right\|^2\right]$$
$$\leq \frac{2\tilde{\eta}^2}{MK}\sum_{m,j}\mathbb{E}_t\left[\left\|\nabla f_m\left(w_t^{(m,j)}; x_t^{(m,j)}\right) - \nabla f_m\left(w_t; x_t^{(m,j)}\right)\right\|^2\right]$$
$$+ \frac{2\tilde{\eta}^2}{MK}\sum_{m,j}\mathbb{E}_t\left[\left\|\nabla f_m\left(w_t; x_t^{(m,j)}\right)\right\|^2\right]$$
$$\overset{(i)}{\leq} 2L^2\eta^2\xi_t + \frac{4\tilde{\eta}^2}{MK}\sum_{m,j}\mathbb{E}_t\left[\left\|\nabla f_m\left(w_t; x_t^{(m,j)}\right) - \nabla F_m(w_t)\right\|^2\right]$$
$$+ \frac{4\tilde{\eta}^2}{M}\sum_{m}\mathbb{E}_t\left[\|\nabla F_m(w_t)\|^2\right]$$
$$\overset{(ii)}{\leq} 2L^2\tilde{\eta}^2\xi_t + 4\tilde{\eta}^2\sigma^2 + \frac{4\tilde{\eta}^2}{M}\sum_{m}\mathbb{E}_t\left[\|\nabla F_m(w_t)\|^2\right].$$

In $(i)$ we use the sample-wise smoothness (Assumption 4.2) and the triangle inequality, and in $(ii)$ we use the uniform bound on the noise of the gradient estimator (Assumption 4.3).

Therefore, we have:

$$\xi_t \leq \frac{4\tilde{\eta}^2}{1 - 2L^2\tilde{\eta}^2} \left( \sigma^2 + \frac{1}{M} \sum_m \mathbb{E}_t \left[ \|\nabla F_m(w_t)\|^2 \right] \right). \tag{18}$$

Plugging (17) and (18) back into (16), together with Lemma D.1 and the condition on the step size then yields

$$\mathbb{E}_t[F(w_{t+1}) - F^*] \leq \mathbb{E}_t[F(w_t) - F^*] - \frac{\tilde{\eta}}{2}\mathbb{E}_t\left[\|\nabla F(w_t)\|^2\right] + \frac{4\tilde{\eta}C_\infty\sigma^2}{\nu_{ps}MK}$$
$$+ \frac{40\tilde{\eta}^2 L}{98\delta^2} \left( \sigma^2 + \frac{1}{M} \sum_m \mathbb{E}_t\left[\|\nabla F_m(w_t)\|^2\right] \right).$$

By taking the full expectation and using the heterogeneity assumption 4.4, we have that:

$$\Delta_{t+1} \leq \Delta_t - \frac{\tilde{\eta}}{2}\left(1 - \frac{80}{98}\tilde{\eta}L\delta^2\right)\mathbb{E}\left[\|\nabla F(w_t)\|^2\right] + \frac{4\tilde{\eta}C_\infty\sigma^2}{\nu_{ps}MK} \tag{19}$$
$$+ \frac{40\tilde{\eta}^2 L}{98}\left(\frac{\theta^2 + \sigma^2}{\delta^2}\right)$$

Using the condition on the local step size on the second term in the RHS of (19) proves the lemma. $\quad\square$

We now prove Theorem 4.7.

**Theorem D.6** (Theorem 4.7 in the main paper). *For the problem class $\mathcal{F}_3(L, \sigma, \theta, \delta, \tau)$, if the constant step size satisfies*

$$\eta \leq \mathcal{O}\left(\frac{1}{LK\delta^2}\right) < \frac{1}{L},$$

*then the iterates of Local SGD satisfy*

$$\mathbb{E}\left[\|\nabla F(\hat{w}_T)\|^2\right] \leq \mathcal{O}\left(\frac{\Delta_0}{\eta KT} + \frac{C_\infty\sigma^2}{\nu_{ps}MK} + \frac{L\eta K(\theta^2 + \sigma^2)}{\delta^2}\right).$$

*Proof.* By rearranging the terms in Lemma D.5 and taking the average from $t = 0$ to $T - 1$:

$$\frac{1}{T}\sum_{t=0}^{T-1}\mathbb{E}\left[\|\nabla F(w_t)\|^2\right] \leq \frac{98}{45}\frac{\Delta_0}{\tilde{\eta}T} + \frac{392}{45}\frac{C_\infty\sigma^2}{\nu_{ps}MK} + \frac{40}{45}\frac{L\tilde{\eta}(\theta^2 + \sigma^2)}{\delta^2}. \tag{20}$$

We conclude the proof by replacing $\tilde{\eta} = \eta K$, and noticing that with $\hat{w}_T$ sampled uniformly from $w_0, \ldots w_{T-1}$, we have

$$\frac{1}{T}\sum_{t=0}^{T-1}\mathbb{E}\left[\|\nabla F(w_t)\|^2\right] = \mathbb{E}\left[\|\nabla F(\hat{w}_T)\|^2\right].$$

$\square$

We now establish Corollary 4.8, which characterizes the communication and sample complexities of Local SGD.

**Corollary D.7** (Corollary 4.8 in the main paper). *For the problem class $\mathcal{F}_3(L, \sigma, \theta, \delta, \tau)$, with step size*

$$\eta \leq \mathcal{O}\left(\frac{\delta^2\epsilon^2}{KL(\theta^2 + \sigma^2)} \wedge \frac{1}{KL\delta^2}\right),$$

*the required number of local steps and communication rounds for Local SGD to achieve $\mathbb{E}[\|\nabla F(\hat{w}_T)\|^2] \leq \epsilon^2$ are given by:*

$$K = \mathcal{O}\left(\frac{\tau\sigma^2}{M\epsilon^2}\right), \quad T = \mathcal{O}\left(\frac{L\Delta_0}{\epsilon^2}\left(\delta^2 \vee \frac{\theta^2 + \sigma^2}{\delta^2\epsilon^2}\right)\right).$$

*Proof.* By choosing

$$K \geq \frac{392}{15} \frac{C_\infty \sigma^2}{\nu_{ps} M \epsilon^2},$$

$$\eta \leq \left( \frac{3}{8} \frac{\delta^2 \epsilon^2}{KL(\theta^2 + \sigma^2)} \wedge \frac{1}{10KL\delta^2} \right),$$

$$T \geq \frac{98}{15} \frac{\Delta_0}{\eta K \epsilon^2} \geq \frac{98}{15} \frac{L\Delta_0}{\epsilon^2} \left( 10\delta^2 \vee \frac{8}{3} \frac{\theta^2 + \sigma^2}{\delta^2 \epsilon^2} \right),$$

(20) guarantees that $\mathbb{E}\left[ \|\nabla F(\hat{w}_T)\|^2 \right] \leq \epsilon^2$.

$\square$

## D.3 Local SGD with Momentum

All the proofs in this section are for the problem class $\mathcal{F}_2(L, \sigma, \tau)$. Recall that the update rule of Local SGD-M can be written as: $w_{t+1} = w_t - \gamma v_{t+1}$. For the theoretical analysis of Local SGD with momentum, we introduce the following notation:

$$\Xi_t = \mathbb{E}\left[ \|\nabla F(w_t) - v_{t+1}\|^2 \right],$$

$$G_0 = \frac{1}{M} \sum_m \|\nabla F(w_0)\|^2.$$

We begin with the following descent lemma.

**Lemma D.8.** *With global step size*

$$\gamma \leq \frac{1}{L},$$

*we have*

$$\Delta_{t+1} \leq \Delta_t - \frac{\gamma}{2} \mathbb{E}\left[ \|\nabla F(w_t)\|^2 \right] + \frac{\gamma}{2} \Xi_t.$$

*Proof.* Similarly as the proof of Lemma D.2, with the update rule of Local SGD-M: $w_{t+1} = w_t - \gamma v_{t+1}$, we have:

$$\Delta_{t+1} \leq \Delta_t - \frac{\gamma}{2} \mathbb{E}\left[ \|\nabla F(w_t)\|^2 \right] + \frac{\gamma}{2} \Xi_t + \left( \frac{L}{2} - \frac{\gamma}{2} \right) s_t.$$

Applying the condition on $\gamma$ proves the lemma. $\square$

We have the following recursive bound on $\Xi_t$:

**Lemma D.9.** *With condition:*

$$\gamma \leq \frac{1}{\sqrt{60}} \frac{\beta}{L}.$$

*We have:*

$$\Xi_t \leq \left( 1 - \frac{14}{15}\beta \right) \Xi_{t-1} + \frac{\beta}{15} \mathbb{E}\left[ \|\nabla F(w_{t-1})\|^2 \right] + 6\beta \mathbb{E}\left[ \|\bar{g}_t - \nabla F(w_t)\|^2 \right] + 6\beta L^2 \xi_t.$$

*Proof.* We have the following decomposition:

$$\nabla F(w_t) - v_{t+1} = (1 - \beta)(\nabla F(w_t) - v_t) - \beta \left( \frac{1}{MK} \sum_{m,k} \nabla f_m\left( w_t; x_t^{(m,k)} \right) - \nabla F(w_t) \right)$$

$$- \frac{\beta}{MK} \sum_{m,k} \left( \nabla f_m\left( w_t^{(m,k)}; x_t^{(m,k)} \right) - \nabla f_m\left( w_t; x_t^{(m,k)} \right) \right)$$

$$= (1 - \beta)(\nabla F(w_t) - v_t) - \beta(\bar{g}_t - \nabla F(w_t)) - \beta(g_t - \bar{g}_t).$$

Hence, by using Young's inequality:

$$(a + b)^2 \leq \left(1 + \frac{\beta}{2}\right) a^2 + \left(1 + \frac{2}{\beta}\right) b^2.$$

We have:

$$
\begin{aligned}
\Xi_t \leq & \left(1 + \frac{\beta}{2}\right)(1 - \beta)^2 \,\mathbb{E}\left[\|\nabla F(w_t) - v_t\|^2\right] \\
& + \left(1 + \frac{2}{\beta}\right)\beta^2 \mathbb{E}\left[\|(g_t - \bar{g}_t) - (\bar{g}_t - \nabla F(w_t))\|^2\right] \\
\leq & \left(1 + \frac{\beta}{2}\right)(1 - \beta)^2 \,\mathbb{E}\left[\|\nabla F(w_t) - v_t\|^2\right] + 2\left(1 + \frac{2}{\beta}\right)\beta^2 \mathbb{E}\left[\|\bar{g}_t - \nabla F(w_t)\|^2\right] \\
& + 2\left(1 + \frac{2}{\beta}\right)\beta^2 L^2 \xi_t.
\end{aligned}
$$

Here, in the last step, we use the triangle inequality and the sample-wise smoothness (Assumption 4.2). Now, since we have:

$$
\begin{aligned}
\nabla F(w_t) - v_t &= \nabla F(w_{t-1}) - v_t + (\nabla F(w_t) - \nabla F(w_{t-1})) \\
&= \Xi_{t-1} + (\nabla F(w_t) - \nabla F(w_{t-1})).
\end{aligned}
$$

Using again Young's inequality, we have:

$$
\begin{aligned}
\Xi_t \leq & (1 + \frac{\beta}{2})(1 - \beta)^2 \left[\left(1 + \frac{\beta}{2}\right)\Xi_{t-1} + \left(1 + \frac{2}{\beta}\right)\mathbb{E}\left[\|\nabla F(w_t) - \nabla F(w_{t-1})\|^2\right]\right] \\
& + 6\beta\mathbb{E}\left[\|\bar{g}_t - \nabla F(w_t)\|^2\right] + 6\beta L^2 \xi_t \\
\leq & (1 - \beta)\Xi_{t-1} + \frac{2L^2}{\beta}\mathbb{E}\left[\|w_t - w_{t-1}\|^2\right] + 6\beta\mathbb{E}\left[\|\bar{g}_t - \nabla F(w_t)\|^2\right] + 6\beta L^2 \xi_t.
\end{aligned}
\tag{21}
$$

Here since $\beta \leq 1$, in the first inequality we use $1 + \frac{2}{\beta} \leq \frac{3}{\beta}$; in the second inequality we use $\left(1 + \frac{\beta}{2}\right)^2 (1 - \beta)^2 \leq (1 - \beta)$ and $\left(1 + \frac{\beta}{2}\right)\left(1 + \frac{2}{\beta}\right)(1 - \beta) \leq \frac{2}{\beta}$ together with the smoothness of $F$.

Furthermore, by the update rule, we have:

$$
\begin{aligned}
\mathbb{E}\left[\|w_t - w_{t-1}\|^2\right] &= \gamma^2 \mathbb{E}\left[\|v_{t-1}\|^2\right] \\
&\leq 2\gamma^2 \Xi_{t-1} + 2\gamma^2 \mathbb{E}\left[\|\nabla F(w_{t-1})\|^2\right].
\end{aligned}
\tag{22}
$$

Plugging (22) back into (21) we have:

$$\Xi_t \leq \left(1 - \beta + \frac{4\gamma^2 L^2}{\beta}\right)\Xi_{t-1} + \frac{4\gamma^2 L^2}{\beta}\mathbb{E}\left[\|\nabla F(w_{t-1})\|^2\right] + 6\beta\mathbb{E}\left[\|\bar{g}_t - \nabla F(w_t)\|^2\right] + 6\beta L^2 \xi_t.$$

We conclude the proof by using the condition on the global step size $\gamma$. $\qquad\square$

Next, we continue to bound the drift $\xi_t$.

**Lemma D.10.** *With the following conditions:*

$$\eta K L \leq \frac{1}{2\beta}.$$

*We have:*

$$\xi_t \leq \; 16 \, (\eta\beta K)^2 \, \sigma^2 + 32 \, [\eta \, (1-\beta) \, K]^2 \, \Xi_{t-1} + 32 \, [\eta \, (1-\beta) \, K]^2 \, \mathbb{E}\left[\|\nabla F \, (w_{t-1})\|^2\right]$$
$$+ \, 16 \, (\eta\beta K)^2 \, \frac{1}{M} \sum_m \mathbb{E}\left[\|\nabla F_m \, (w_t)\|^2\right].$$

*Proof.* We start by using Young inequality $(a+b)^2 \leq \left(1 + \frac{1}{K-1}\right) a^2 + K b^2$. Then, for any $0 \leq k \leq K-1$ we have:

$$\mathbb{E}\left[\left\|w_t^{(m,k)} - w_t\right\|^2\right] \leq \; \left(1 + \frac{1}{K-1}\right) \mathbb{E}\left[\left\|w_t^{(m,k-1)} - w_t\right\|^2\right] + K\mathbb{E}\left[\left\|w_t^{(m,k-1)} - w_t^{(m,k)}\right\|^2\right]$$
$$= \; \left(1 + \frac{1}{K-1}\right) \mathbb{E}\left[\left\|w_t^{(m,k-1)} - w_t\right\|^2\right]$$
$$+ \, K\eta^2 \mathbb{E}\left[\left\|\beta\nabla f_m\left(w_t^{(m,k-1)}; x_t^{(m,k-1)}\right) + (1-\beta) \, v_t\right\|^2\right].$$

Now we have the following decomposition:

$$\beta\nabla f_m\left(w_t^{(m,k-1)}; x_t^{(m,k-1)}\right) + (1-\beta) \, v_t = (1-\beta) \, v_t + \beta\nabla F_m\left(w_t\right)$$
$$+ \, \beta\left[\nabla f_m\left(w_t^{(m,k-1)}; x_t^{(m,k-1)}\right) - \nabla f_m\left(w_t; x_t^{(m,k-1)}\right)\right]$$
$$+ \, \beta\left[\nabla f_m\left(w_t; x_t^{(m,k-1)}\right) - \nabla F_m\left(w_t\right)\right].$$

Hence, by using the triangle inequality, we have:

$$\mathbb{E}\left[\left\|w_t^{(m,k)} - w_t\right\|^2\right] \leq \left(1 + \frac{1}{K-1}\right) \mathbb{E}\left[\left\|w_t^{(m,k-1)} - w_t\right\|^2\right]$$
$$+ \, 4\eta^2\beta^2 K\mathbb{E}\left[\left\|\nabla f_m\left(w_t^{(m,k-1)}; x_t^{(m,k-1)}\right) - \nabla f_m\left(w_t; x_t^{(m,k-1)}\right)\right\|^2\right]$$
$$+ \, 4\eta^2\beta^2 K\mathbb{E}\left[\left\|\nabla f_m\left(w_t; x_t^{(m,k-1)}\right) - \nabla F_m\left(w_t\right)\right\|^2\right]$$
$$+ \, 4\eta^2\beta^2 \mathbb{E}\left[\|\nabla F(w_t)\|^2\right]$$
$$+ \, 8\eta^2 \, (1-\beta)^2 \, K\left(\Xi_{t-1} + \mathbb{E}\left[\|\nabla F \, (w_{t-1})\|^2\right]\right)$$
$$\leq \left(1 + \frac{1}{K-1} + 4\eta^2\beta^2 L^2 K\right) \mathbb{E}\left[\left\|w_t^{(m,k-1)} - w_t\right\|^2\right] + 4K\mathcal{A},$$

with

$$\mathcal{A} = \eta^2\beta^2 \left(\sigma^2 + \mathbb{E}\left[\|\nabla F_m \, (w_t)\|^2\right]\right) + 2\eta^2 \, (1-\beta)^2 \left(\Xi_{t-1} + \mathbb{E}\left[\|\nabla F \, (w_{t-1})\|^2\right]\right).$$

Here, we use the sample-wise smoothness and the uniform bound on the gradient noise in the last step (Assumptions 4.2, 4.3).

Unrolling the above recursion, with $w_t^{(m,0)} = w_t$, we have:

$$\mathbb{E}\left[\left\|w_t^{(m,k)} - w_t\right\|^2\right] \leq \; 4K\mathcal{A}\sum_{j=0}^{k-1}\left(1 + \frac{1}{K-1} + 4\eta^2\beta^2 L^2 K\right)^j.$$

With the condition on $\eta$, we have that $4\eta^2\beta^2 L^2 K \leq \frac{1}{K} < \frac{1}{K-1}$. Hence, by using the geometric sum formula, we have:

$$\sum_{j=0}^{k-1} \left(1 + \frac{1}{K-1} + 4\eta^2\beta^2 L^2 K\right)^j \leq \frac{1 - \left(1 + \frac{2}{K-1}\right)^k}{-\frac{2}{K-1}}$$

$$= \frac{K-1}{2}\left[\left(1 + \frac{2}{K-1}\right)^k - 1\right]$$

$$\leq \frac{K-1}{2}\left(1 + \frac{2}{K-1}\right)^{K-1}$$

$$\leq \frac{e^2}{2}(K-1) \leq 4K.$$

We conclude the lemma by taking the average over $K$ and $M$. $\qquad\square$

In the proof of Local SGD, we have to use the heterogeneity assumption to bound the term $\frac{1}{M}\sum_m \mathbb{E}\left[\|\nabla F_m(w_t)\|^2\right]$. The use of momentum allows us to build a recursive bound for this term by the following lemma.

**Lemma D.11.** *We have that:*

$$\frac{1}{M}\sum_m \mathbb{E}\left[\|\nabla F_m(w_t)\|^2\right] \leq 3G_0 + 6(1+t)\gamma^2 L^2 \sum_{j=0}^{t-1}\left(\Xi_j + \mathbb{E}\left[\|\nabla F(w_j)\|^2\right]\right).$$

*Proof.* By using Young's inequality and the smoothness of $F_m$, we have that:

$$\mathbb{E}\left[\|\nabla F_m(w_t)\|^2\right] \leq (1+a)\mathbb{E}\left[\|\nabla F_m(w_{t-1})\|^2\right] + \left(1 + a^{-1}\right)L^2\mathbb{E}\left[\|w_t - w_{t-1}\|^2\right]$$

$$\leq (1+a)\mathbb{E}\left[\|\nabla F_m(w_{t-1})\|^2\right] + 2\left(1 + a^{-1}\right)\gamma^2 L^2\left(\Xi_{t-1} + \mathbb{E}\left[\|\nabla F(w_{t-1})\|^2\right]\right).$$

By unrolling, we have:

$$\mathbb{E}\left[\|\nabla F_m(w_t)\|^2\right] \leq (1+a)^t\mathbb{E}\left[\|\nabla F_m(w_0)\|^2\right] + 2\left(1 + a^{-1}\right)\gamma^2 L^2\sum_{j=0}^{t-1}\left(\Xi_j + \mathbb{E}\left[\|\nabla F(w_j)\|^2\right]\right)(1+a)^{t-1-j}$$

$$\leq e^{at}\mathbb{E}\left[\|\nabla F_m(w_0)\|^2\right] + 2e^{at}\left(1 + a^{-1}\right)\gamma^2 L^2\sum_{j=0}^{t-1}\left(\Xi_j + \mathbb{E}\left[\|\nabla F(w_j)\|^2\right]\right).$$

By taking $a = t^{-1}$ and taking the average over $m \in [M]$, we conclude the lemma. $\qquad\square$

**Lemma D.12.** *With the following conditions:*

$$\eta KL \leq \left(\frac{1}{2\beta} \wedge \sqrt{\frac{C_\infty}{120\nu_{ps}MK\beta^2}} \wedge \sqrt{\frac{L\Delta_0}{360\beta^3 TG_0}} \wedge \frac{1}{225}\frac{1}{(1-\beta)} \wedge \frac{1}{525}\frac{1}{\beta\gamma LT}\right),$$

$$\gamma \leq \frac{1}{\sqrt{60}}\frac{\beta}{L}.$$

*We have that:*

$$\frac{1}{T}\sum_{t=0}^{T-1}\Xi_t \leq \frac{7}{2}\frac{L\Delta_0}{\beta T} + 31\frac{C_\infty\sigma^2}{\nu_{ps}MK} + \frac{1}{4}\sum_{t=0}^{T-1}\mathbb{E}\left[\|\nabla F(w_t)\|^2\right].$$

*Proof.* By Lemma D.11, we have that:

$$\frac{1}{M}\sum_{t=0}^{T-1}\sum_{m}\mathbb{E}\left[\|\nabla F_m\left(w_t\right)\|^2\right] \leq 3TG_0 + 6\gamma^2 L^2 \sum_{t=0}^{T-1}(1+t)\sum_{j=0}^{t-1}\left(\Xi_j + \mathbb{E}\left[\|\nabla F\left(w_j\right)\|^2\right]\right)$$

$$\leq 3TG_0 + 12\left(\gamma LT\right)^2 \sum_{t=0}^{T-2}\left(\Xi_t + \mathbb{E}\left[\|\nabla F\left(w_t\right)\|^2\right]\right). \tag{23}$$

Plugging (23) back into Lemma D.10 we have:

$$6\beta L^2 \sum_{t=0}^{T-1}\xi_t \leq 96\beta T\left(\eta KL\beta\right)^2\left(\sigma^2 + 3G_0\right)$$

$$+ \beta \underbrace{\left[192\left(\eta KL\left(1-\beta\right)\right)^2 + 1152\left(\eta KL\beta\right)^2\left(\gamma LT\right)^2\right]}_{\mathcal{B}} \sum_{t=0}^{T-2}\left(\Xi_t + \mathbb{E}\left[\|\nabla F\left(w_t\right)\|^2\right]\right). \tag{24}$$

By condition on the step size, we have that $\mathcal{B} \leq \frac{2}{15}$. Therefore, plugging (24) back into Lemma D.9, by noticing that $\mathbb{E}\left[\|\nabla F\left(w_{-1}\right)\|^2\right] = 0$, gives:

$$\sum_{t=0}^{T-1}\Xi_t \leq \left(1 - \frac{14}{15}\beta\right)\sum_{t=-1}^{T-2}\Xi_t + \frac{\beta}{15}\sum_{t=0}^{T-2}\mathbb{E}\left[\|\nabla F\left(w_t\right)\|^2\right]$$

$$+ 6\beta\sum_{t=0}^{T-1}\mathbb{E}\left[\|\bar{g}_t - \nabla F(w_t)\|\right] + 96\beta T\left(\eta KL\beta\right)^2\left(\sigma^2 + 3G_0\right)$$

$$+ \frac{2\beta}{15}\left(\sum_{t-1}^{T-2}\Xi_t + \sum_{t=0}^{T-2}\mathbb{E}\left[\|\nabla F\left(w_t\right)\|^2\right]\right)$$

$$\leq \left(1 - \frac{4}{5}\beta\right)\sum_{t=-1}^{T-2}\Xi_t + \frac{\beta}{5}\sum_{t=0}^{T-2}\mathbb{E}\left[\|\nabla F\left(w_t\right)\|^2\right]$$

$$+ 6\beta\sum_{t=0}^{T-1}\mathbb{E}\left[\|\bar{g}_t - \nabla F(w_t)\|^2\right] + 96\beta T\left(\eta KL\beta\right)^2\left(\sigma^2 + 3G_0\right).$$

By rearranging the terms, we have:

$$\sum_{t=0}^{T-1}\Xi_t \leq \frac{5}{4\beta}\Xi_{-1} + \frac{1}{4}\sum_{t=0}^{T-1}\mathbb{E}\left[\|\nabla F\left(w_t\right)\|^2\right] + \frac{15}{2}\sum_{t=0}^{T-1}\mathbb{E}\left[\|\bar{g}_t - \nabla F\left(w_t\right)\|^2\right]$$

$$+ 120T\left(\eta KL\beta\right)^2\left(\sigma^2 + 3G_0\right). \tag{25}$$

With the condition on the step size, we have:

$$120(\eta KL\beta)^2\sigma^2 \leq \frac{C_\infty \sigma^2}{\nu_{ps}MK},$$

$$360\left(\eta KL\beta\right)^2 G_0 \leq \frac{L\Delta_0}{\beta T}. \tag{26}$$

And by the law of total expectation, we have:

$$\mathbb{E}\left[\|\bar{g}_t - \nabla F(w_t)\|^2\right] = \mathbb{E}\left[\mathbb{E}_t\left[\|\bar{g}_t - \nabla F(w_t)\|^2\right]\right]$$

$$\leq \frac{4C_\infty \sigma^2}{\nu_{ps}MK}. \quad \text{(Lemma D.1)} \tag{27}$$

Hence, by plugging (26)) and (27) back into (25) we have:

$$\frac{1}{T}\sum_{t=0}^{T-1} \Xi_t \leq \frac{5}{4}\frac{\Xi_{-1}}{\beta T} + \frac{L\Delta_0}{\beta T} + \frac{31 C_\infty \sigma^2}{\nu_{ps} M K} + \frac{1}{4}\sum_{t=0}^{T-1} \mathbb{E}\left[\|\nabla F(w_t)\|^2\right].$$

Finally noticing that if we initialize $v_0 = 0$, $\Xi_{-1} = \mathbb{E}\left[\|\nabla F(w_0)\|^2\right] \leq 2L\Delta_0$ by the smoothness of $F$. Hence, we prove the lemma. $\qquad\square$

Now we are ready to prove Theorem 4.9.

**Theorem D.13** (Theorem 4.9 in the main paper). *For the problem class $\mathcal{F}_2(L, \sigma, \tau)$, with the following conditions on the step sizes:*

$$\eta K L \leq \mathcal{O}\left(\frac{1}{\beta} \wedge \sqrt{\frac{C_\infty}{\nu_{ps} M K \beta^2}} \wedge \sqrt{\frac{L\Delta_0}{\beta^3 T G_0}} \wedge \frac{1}{(1-\beta)} \wedge \frac{1}{\beta \gamma L T}\right),$$

$$\gamma \leq \mathcal{O}\left(\frac{\beta}{L}\right),$$

*the iterates of Local SGD-M satisfy:*

$$\mathbb{E}\left[\|\nabla F(\hat{w}_T)\|^2\right] \leq \mathcal{O}\left(\frac{L\Delta_0}{\beta T} + \frac{C_\infty \sigma^2}{\nu_{ps} M K}\right).$$

*Proof.* By rearranging the terms in Lemma D.8:

$$\frac{1}{T}\sum_{t=0}^{T-1} \mathbb{E}\left[\|\nabla F(w_t)\|^2\right] \leq \frac{2\Delta_0}{\gamma T} + \frac{1}{T}\sum_{t=0}^{T-1} \Xi_t.$$

Using the condition on $\gamma$ and replacing $\frac{1}{T}\sum_{t=0}^{T-1} \Xi_t$ by Lemma D.12 proves the theorem. $\qquad\square$

From Theorem D.13, we can derive directly the communication and sample complexity of Local SGD-M.

**Corollary D.14** (Corollary 4.10 in the main paper). *Under the conditions of Theorem D.13, the required number of local steps and communication rounds for Local SGD-M to achieve $\mathbb{E}[\|\nabla F(\hat{w}_T)\|^2] \leq \epsilon^2$ are:*

$$K = \mathcal{O}\left(\frac{C_\infty \sigma^2}{\nu_{ps} M \epsilon^2}\right), \quad T = \mathcal{O}\left(\frac{L\Delta_0}{\beta \epsilon^2}\right)$$

*Proof.* The proof is a direct consequence of Theorem D.13 and follows the same steps in the proof of Corollary D.4. $\qquad\square$

## D.4 Uniformly Ergodic Markov Processes

A common assumption in the literature on Markov SGD is uniform geometric ergodicity (see Definition B.10).

**Assumption D.15.** The system-level Markov chain is uniformly ergodic. In particular, for any $n \in \mathbb{N}$, we have:

$$\sup_{x \in \Omega} \|P^n(x, .) - \pi\|_{\text{TV}} \leq c\rho^n,$$

for some $c \leq \infty$ and $\rho < 1$.

This assumption, which characterizes exponentially fast convergence to a unique stationary measure of Markov processes, is widely used in the literature of Stochastic Approximation with Markovian noise [3, 37, 15]. We highlight that in the main paper, we only assume that the clients' Markov chains converge to a unique stationary measure, without any further assumption on convergence

speed. In this section, we show that our analysis can be easily extended with this additional stronger assumption.

In particular, from [43, Proposition 3.4], we have that:

$$\frac{1}{\nu_{ps}} \le 2\tau,$$

where $\tau$ is the mixing time of the system-level Markov chain (see Proposition B.11 for definition). According to Proposition B.12, we also have that:

$$\tau \le \lceil \log M + 1 \rceil \tau_{\max},$$

where $\tau_{max} = \max_{m \in [M]} \tau_m$ the maximum mixing time of clients' Markov processes.

Hence, a key implication of the uniform ergodicity assumption is a characterization of the pseudo-spectral gap in terms of the mixing time. That is, $1/\nu_{ps}$ is replaced by $\tau$ in the proof of Lemma D.1. As a consequence, for Minibatch SGD, Local SGD, and Local SGD-M, the new required number of local steps is:

$$K = \tilde{\mathcal{O}}\left(\frac{C_\infty \tau_{\max} \sigma^2}{M \epsilon^2}\right).$$

The communication complexity remains unchanged under the additional assumption of uniform ergodicity.

We note that uniform ergodicity can also be used to characterize $C_\infty$, as is discussed further in Section D.5.

### D.5  Characterization of $C_\infty$

A key quantity in our analysis is $C_\infty$, which impacts the required number of local steps. In particular, recall

$$C_\infty = \sup_{x \in \Omega} \operatorname{ess\,sup} \left| \frac{\mathrm{d}P(x, \cdot)}{\mathrm{d}\pi} \right|.$$

When $|\Omega| < \infty$, $C_\infty$ is given by

$$C_\infty = \max_{x,y \in \Omega} \frac{P(x,y)}{\pi(y)},$$

which can be explicitly computed for the case of two-state reversible Markov chains with $\Omega = \{0, 1\}$ satisfying

$$P(x, y) = \begin{cases} p, & x \ne y \\ 1 - p, & x = y. \end{cases}$$

In this case, $\pi(x) = \frac{1}{2}$, $x \in \{0, 1\}$. It then follows that

$$C_\infty = 2 \max\{1 - p, p\},$$

which is minimized for $p = \frac{1}{2}$, corresponding to independence between consecutive samples from the Markov chain.

For Markov chains with finite state space, where $|\Omega| < \infty$, $C_\infty$ can be bounded using the uniform ergodicity assumption in Assumption D.15. For an arbitrary $x \in \Omega$, under the assumption $P(x, \cdot) \ll \pi$, observe that

$$\begin{aligned}
\frac{1}{2} \sum_{y \in \Omega} |P(x,y) - \pi(y)| &= \frac{1}{2} \sum_{x \in \Omega} \pi(y) \left| \frac{P(x,y)}{\pi(y)} - 1 \right| \\
&\ge \frac{1}{2} \pi_{\min} \left| \max_{y \in \Omega} \frac{P(x,y)}{\pi(y)} - 1 \right| \\
&\ge \frac{1}{2} \pi_{\min} \left( \max_{y \in \Omega} \frac{P(x,y)}{\pi(y)} - 1 \right),
\end{aligned}$$

where $\pi_{\min} = \min_{y \in \Omega} \pi(y)$. It then follows that for arbitrary $x \in \Omega$,

$$\max_{y \in \Omega} \frac{P(x,y)}{\pi(y)} \le \frac{2\|P(x,\cdot) - \pi\|_{\mathrm{TV}}}{\pi_{\min}} + 1.$$

Choosing $x = \arg\max_{u \in \Omega} \max_{y \in \Omega} \frac{P(u,y)}{\pi(y)}$, and applying the uniform ergodicity assumption in Assumption D.15, we then have

$$C_\infty \le \frac{2c\rho}{\pi_{\min}} + 1.$$

For finite Markov chains under the uniform ergodicity assumption, [43, Proposition 3.4] provides a relationship between the total variation distance and the pseudo spectral gap. Applying [43, Proposition 3.4] then yields

$$C_\infty \le \frac{(1 - \gamma_{ps})^{(1 - 1/\gamma_{ps})/2} \sqrt{\frac{1}{\pi_{\min}} - 1}}{\pi_{\min}} + 1$$
$$\le \frac{e\sqrt{1/\pi_{\min} - 1}}{\pi_{\min}} + 1$$

### D.6 Alternative Bounds for Stochastic Gradient Errors in the Non-Stationary Case

In Lemma D.1, we obtained a bound on the stochastic gradient error in terms of $C_\infty$ by choosing the stationary distribution as a reference. In the following lemma, we obtain an alternative bound on the stochastic gradient error by choosing a different reference distribution. This bound can distinguish between stationary and non-stationary Markovian data processes.

**Lemma D.16.** *Let $\mu$ be the initial distribution of the system-level Markov chain defined in Section 3.2. We assume that for every $t \in \mathbb{N}$, and for every $x \in \Omega$, $P(x,.)$ is absolutely continuous with respect to $\mu P^{Kt}$, and we define:*

$$C_{q,t} = \sup_{x \in \Omega} \operatorname{ess\,sup} \left| \frac{dP(x,.)}{d(qP^{kt})} \right| < \infty \tag{28}$$

*If Assumptions 3.1 and 4.3 hold, then we have:*

$$\mathbb{E}_t \left[ \left\| \frac{1}{MK} \sum_{m,k} \nabla f_m \left( w_t; x_t^{(m,k)} \right) - \nabla F(w_t) \right\|^2 \right] \le \frac{4C_{q,t}\sigma^2}{\nu_{ps} MK} + 2C_{q,t}\sigma^2 \left\| qP^{Kt} - \pi \right\|_{\mathrm{TV}}.$$

*Proof.* Let $\rho$ be a distribution such that $\mu P^{Kt} \ll \rho$ and $\pi \ll \rho$. We then have:

$$\mathbb{E}_t \left[ \left\| \frac{1}{MK} \sum_{m,k} \nabla f_m \left( w_t; x_t^{(m,k)} \right) - \nabla F(w_t) \right\|^2 \right]$$

$$= \mathbb{E}_{qP^{kt}} \left[ \frac{dP\left( x_{t-1}^{K-1}, . \right)}{d(qP^{kt})} \left\| \frac{1}{MK} \sum_{m,k} \nabla f_m \left( w_t; x_t^{(m,k)} \right) - \nabla F(w_t) \right\|^2 \right]$$

$$\leq \left\| \frac{dP(x_{t-1}^{K-1}, .)}{d\left( qP^{Kt} \right)} \right\|_{\pi,\infty} \mathbb{E}_{qP^{Kt}} \left[ \left\| \frac{1}{MK} \sum_{m,k} \nabla f_m \left( w_t; x_t^{(m,k)} \right) - \nabla F(w_t) \right\|^2 \right] \quad \text{(Holder's inequality)}$$

$$\leq C_{q,t} \mathbb{E}_{qP^{Kt}} \left[ \left\| \frac{1}{MK} \sum_{m,k} \nabla f_m \left( w_t; x_t^{(m,k)} \right) - \nabla F(w_t) \right\|^2 \right] \quad \text{(By using (28))}$$

$$\leq C_{q,t} \mathbb{E}_\pi \left[ \left\| \frac{1}{MK} \sum_{m,k} \nabla f_m \left( w_t; x_t^{(m,k)} \right) - \nabla F(w_t) \right\|^2 \right]$$

$$+ C_{q,t} \int_\Omega \left\| \frac{1}{MK} \sum_{m,k} \nabla f_m \left( w_t; x_t^{(m,k)} \right) - \nabla F(w_t) \right\|^2 \left| \frac{d\left( qP^{Kt} \right)}{d\rho} - \frac{d\pi}{d\rho} \right| d\rho$$

$$\leq \frac{4C_{q,t}\sigma^2}{\nu_{ps}MK} + 2C_{q,t}\sigma^2 \left\| qP^{Kt} - \pi \right\|_{\text{TV}}$$

In the last step, we follow (11) to bound the squared error of the gradient estimator at the stationary measure and Assumption 4.3 together with the definition of the total variation norm (Appendix B,Definition B.7). $\qquad \square$

Observe that in the stationary case, for every $t \in \mathbb{N}$ the additional term $\|qP^{Kt} - \pi\|_{\text{TV}}$ is zero, and $C_{q,t} = C_\infty$, and we recover the bound in Lemma D.1. On the other hand, in the non-stationary case, while the total variation norm can be bound using [43, Proposition 3.4], obtaining a tighter bound than the one in Lemma D.1 is challenging due to the term $C_{q,t}$.

# E   Additional Experiments

## E.1   Experimental Setting

**Computing environment:** All the experiments performed in this paper are run entirely on many different CPU clusters provided by the Grid'5000 testbed, with different types of CPU (e.g., Intel Xeon E5-2698, Intel Xeon E5-2620, Intel Xeon E5-2630). All the software packages and datasets used for experiments in this paper are open-sourced, with the exact version provided. For the main experiments, we use 10 different random seeds, and report the average together with the 95% confidence interval. Further detailed instructions to run the experiments are provided in the supplementary material.

In Section 5, we evaluated the performance of Local SGD, Minibatch SGD, and Local SGD with Momentum on the Beijing Multi-Site Air-Quality dataset. In this section, we provide details on the data preprocessing and the learning problem.

The Beijing Multi-Site Air-Quality dataset consists of hourly measurements from 12 weather stations across China, collected between March 2013 and February 2017. In particular, the dataset consists of measurements of 6 pollutants (PM2.5, PM10, S02, CO2, CO, and O3) and 6 meteorological indicators (temperature, pressure, dew point, rainfall, wind speed, and wind direction). In total, there are more than 400,000 measurements of each variable.

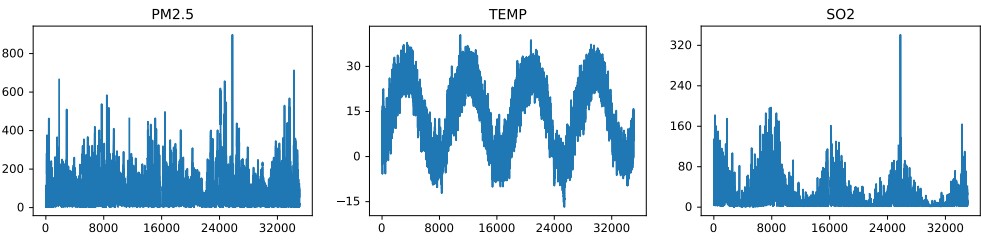

Figure 3: Original time series for PM2.5 pollution, temperature, and SO2 pollution. Observe the periodicity in the data indicating seasonality.

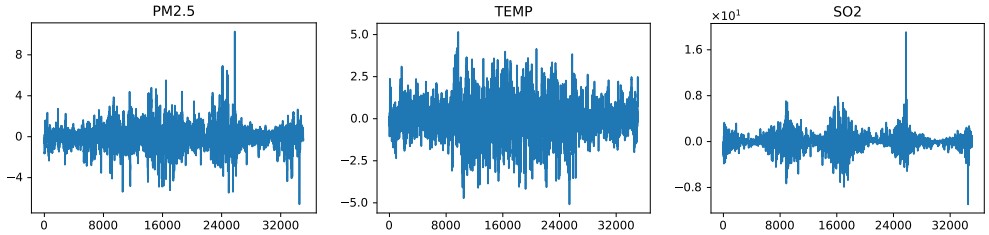

Figure 4: Time series of PM2.5, temperature, and SO2 with seasonality removed, which is utilized (along with the other pollution and meteorological variables) for the experiments.

As illustrated in Figure 3, the dataset has a strong seasonality effect. As we seek to experimentally validate our theoretical analysis, which requires convergence to a stationary distribution. As such, we estimate and remove the seasonality via the STATSMODEL python package [52]. We then drop the PM10 and WD (wind direction) features, fill in the missing values with their average values during the last month, and normalize the data. In Figure 4, we plot some measures after these steps of preprocessing.

The focus of our theoretical analysis is on Markovian data. Pollution and meteorological data are commonly modelled via Markov chains. To verify that a Markovian model is relevant to the dataset, it is necessary to verify that there is dependence between samples. To this end, in Fig. 5 we plot the autocorrelation of the PM2.5 concentration time series. Observe that while the autocorrelation decreases as the lag increases, which indicates mixing. On the other hand, the autocorrelation with a

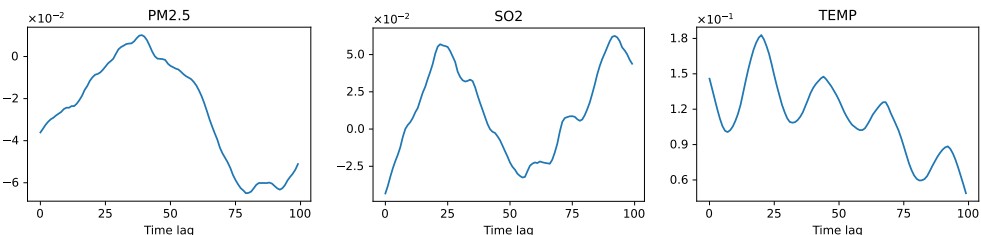

Figure 5: Auto-correlation of PM2.5, SO2, and temperature time series with seasonality removed. Observe that, particularly for the temperature, there are high levels of correlation at small lags, indicating dependence between adjacent samples.

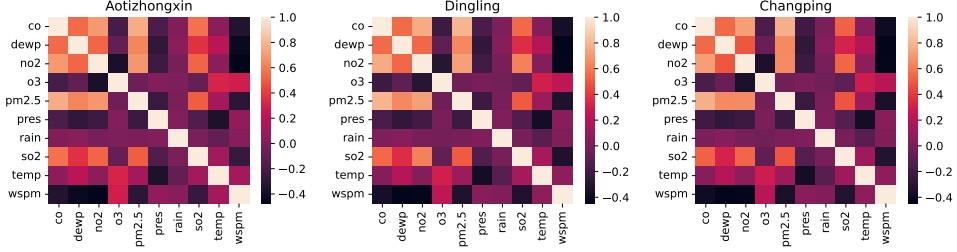

Figure 6: Correlation matrix of the data at three stations.

small lag remains high, indicating the presence of dependence and the need for non-independent and, in particular, Markovian models for the evolution of the PM2.5 concentration.

The learning problem we consider is linear regression with the PM2.5 measurements as the response, and the meteorological variables and the other pollution variables utilized as covariates. From the correlation matrices in Figure 6, we can see that there is a strong correlation between PM2.5 and CO, DEWP (dew point), and NO2, while other features are less relevant. Hence, we consider the following loss function:

$$f\left(w;(x,y)\right) = \left(w^T x - y\right)^2 + \lambda r(w),$$
$$r(w) = \frac{1}{2} \sum_{i=1}^{d=9} \frac{w_i^2}{1 + w_i^2}.$$

where $y \in \mathbb{R}$ is the PM2.5 measure, $x \in \mathbb{R}^9$ is the vector of other variables measured at the same hour, and $w \in \mathbb{R}^d$ is the model parameter. The non-convex regularizer encourages small weights for irrelevant features.

To allocate the data over different clients, for each client we randomly sample $n \in \{6, 12\}$ consecutive months from the first 36 months from all the sites with seasonality removal. The remaining 12 months are reserved for testing.

### E.2 Step Size Scaling with the Number of Local Steps

We remind here the upper bounds for Local SGD in Theorem 4.7:

$$\mathbb{E}\left[\|\nabla F(\hat{w}_T)\|^2\right] \leq \mathcal{O}\left(\frac{\Delta_0}{\eta KT} + \frac{C_\infty \sigma^2}{\nu_{ps} MK} + \frac{LK\eta\left(\theta^2 + \sigma^2\right)}{\delta^2}\right).$$

In Section 5, we perform experiments with a fixed local step size $\eta$ to better illustrate the heterogeneity effect on Local SGD when $K$ increases. In this section, we perform the same experiments as in Figure 7, but with the local step size $\eta$ scaled as $b/K$, where $b \in \{0.1, 0.01\}$. This replaces the fixed

Table 2: Mean gradient norm of Local SGD

| $\eta$ | $K = 10$ | $K = 50$ | $K = 100$ |
|---|---|---|---|
| $0.1/K$ | $0.00506 \pm 0.00076$ | $0.00307 \pm 0.00041$ | $0.00233 \pm 0.00030$ |
| $0.01/K$ | $0.01248 \pm 0.00279$ | $0.00921 \pm 0.00210$ | $0.01190 \pm 0.00245$ |

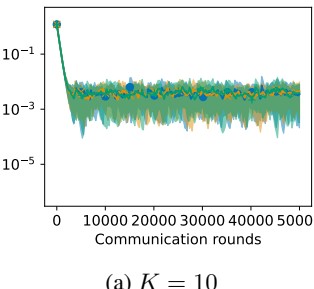 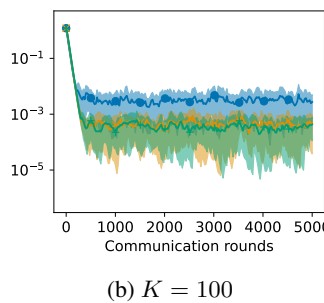 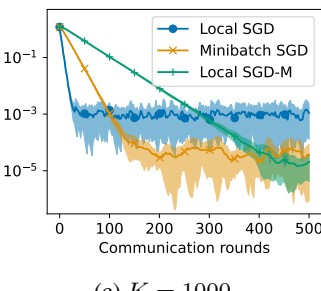

(a) $K = 10$        (b) $K = 100$        (c) $K = 1000$

Figure 7: Gradient norm as a function of the number of communication rounds for Local SGD, Minibatch SGD, and Local SGD-M, with 10 clients, $\gamma = 0.01, \eta = 0.001, \beta = 0.1$, and the mixing time for every client $\tau_m = 100$.

local step size $\eta = 0.01$ in the original experiments. We report the results after 10 different seeds together with the 95% confidence interval in Table 2.

From the above upper bound, we note that, for moderate values of the initial suboptimality gap and heterogeneity:

- If $K$ is large, then the bound is dominated by the heterogeneity term.
- If $K$ is small, then the bound is dominated by the variance term.

Choosing the step size as $\eta \propto 1/K$ leaves the first and third error terms unchanged with $K$, while the second term decays with increasing $K$. Hence, when this second term dominates for small $K$, the gradient norm $||\nabla F(w)||^2$ should decrease before flattening out at a positive plateau. The patterns shown in each row align with this qualitative prediction.

The comparison of the two rows also supports the fact that the "variance" term cannot be controlled by the step size, which was discussed in Appendix C. Indeed, for a fixed $K$, reducing the step size by a factor of 10 does not improve the performance. On the contrary, the gradient norm appears to increase for smaller $\eta$. This can be explained by the first term in the bound, associated with the suboptimality gap, which scales inversely with the step size.

### E.3 Additional Experiments on Synthetic Data

In this section, we experimentally evaluate the performance of Minibatch SGD, local SGD, and local SGD-M on the linear regression problem with non-convex regularization with synthetic Markovian data. In the following experiments, we study the average performance over 10 random seeds in a setting similar to [15].

Let $\mathcal{U}(a, b)$ denote the uniform probability distribution over $[a, b]$. The data stream for each client $m$ is generated by a two-state Markov chain $(i_t^{(m,k)}, k \in [K], t \in \mathbb{N})$, where the probability of jumping from one state to another is $p \in (0, 1)$, with mixing time $\Theta(1/p)$. We note that for reversible Markov chains, as considered in these experiments, the mixing time is related to the spectral gap via

$$\frac{1}{\nu_{ps}} \leq 2\tau_{mix}$$

Associated with each state $i \in \{0, 1\}$ are the vectors $V_{m,i} \in \mathbb{R}^{10}$ which are drawn randomly for each seed according to $\mathcal{U}(0, 1)$. For each seed and $i \in \{0, 1\}$, half of the optimal parameters $w_{m,i} \in \mathbb{R}^{10}$ take the value $w_i^1 \sim \mathcal{U}(0, 1)$, while the other half take the value $w_i^2 \sim \mathcal{U}(1, 2)$.

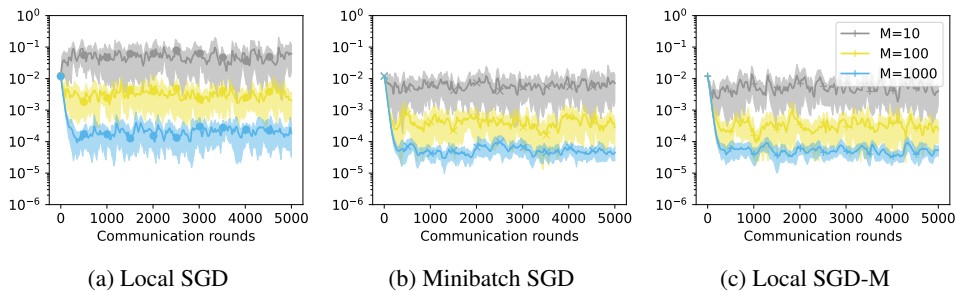

(a) Local SGD     (b) Minibatch SGD     (c) Local SGD-M

Figure 8: Gradient norm as a function of the number of communication rounds for Local SGD, Minibatch SGD & Local SGD-M, with $\gamma = 0.01, \eta = 0.001, \beta = 0.1, \lambda = 0.01, \tau_m = 100$ for all $m$ and $K = 100$.

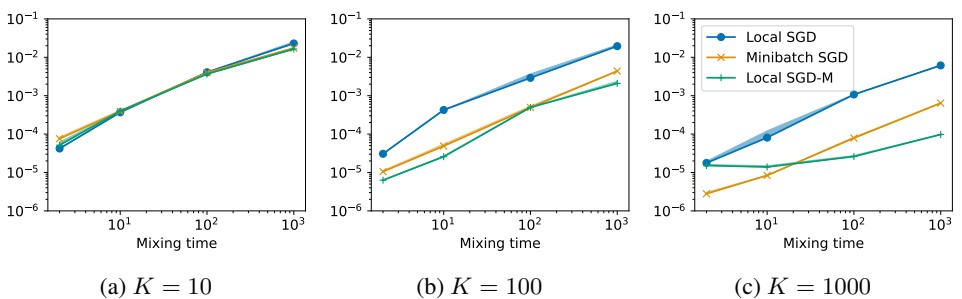

(a) $K = 10$     (b) $K = 100$     (c) $K = 1000$

Figure 9: The gradient norm of the last iterate of 3 algorithms as a function of the mixing time with $\gamma = 0.01, \eta = 0.001, \beta = 0.1, \lambda = 0.01$, 100 clients and different numbers of local steps.

The samples associated with client $m$ correspond to the observations

$$x_t^{(m,k)} = w_{m,i_t^{(m,k)}}^T V_{m,i_t^{(m,k)}} + \epsilon_t^{(m,k)},$$

where $\epsilon_t^{(m,k)} \sim \mathcal{U}(0, 0.01)$, $t \in \mathbb{N}, m \in [M], k \in [K]$.

Since the Markov chain $(i_t^{(m,k)}, \ k \in [K], t \in \mathbb{N})$ is symmetric, the stationary distribution for every client $m$ is uniform; i.e., $\pi_m(0) = \pi_m(1) = 1/2$. The local objective function for each client $m$ is then given by

$$
\begin{aligned}
F_m(w) = &\frac{1}{2}\left[(w - w_{m,1})^T V_{m,1}\right]^2 \\
&+ \frac{1}{2}\left[(w - w_{m,2})^T V_{m,2}\right]^2 + \lambda r(w),
\end{aligned}
$$

where $r(w) = \frac{1}{2}\sum_{d=1}^{10} \frac{w_{[d]}^2}{1+w_{[d]}^2}$ is a non-convex regularizer often used in robust non-convex and smooth stochastic optimization [57, 20].

**Impact of Heterogeneity and Number of Clients**: Figure 7 and 8 confirm that synthetic data experiments replicate the trends observed on real-world data: Local SGD's performance degrades under heterogeneity, while all three algorithms improve with more clients—consistently supporting our theoretical analysis.

**Effect of mixing time**: In Figure 9, we plot the gradient norm of the last iterate of Minibatch SGD, Local SGD, and Local SGD-M as a function of the mixing time. As predicted by our analysis, the performance of Minibatch SGD, Local SGD, and Local SGD-M degrades when the mixing time increases.

# F   Limitations

**Heterogeneity and Smoothness Assumptions:** In our work, we have imposed the bounded gradient dissimilarity (BGD) assumption. As noted in Section 4.1, this assumption was first introduced in [27] and has since become ubiquitous in the analysis of local SGD [64]. We highlight that the BGD assumption includes the assumptions in [60, 33]. While weaker heterogeneity assumptions exist in the literature, these are not straightforward to apply for non-convex objectives where the BGD assumption is standard [30].

**Statistical Assumptions:** The focus of this work is on time series data with temporal dependence modeled via Markov chains. These models are widely used in many areas of statistical modeling, ranging from physical and biological systems to queuing systems. On the other hand, Markovian models are not the only common time series models, and, for example, do not include all autoregressive models.

We also assumed that the stochastic gradient estimates have bounded errors. Nevertheless, this is a common assumption in stochastic optimization with bounded noise.

**Optimality:** At present, tight lower bounds for Markovian data are not known. As such, we were not able to establish the optimality of our algorithm. Instead, we have compared with the lower bound in the i.i.d. setting. We are only able to assert that in this setting, Minibatch SGD and Local SGD with momentum can at least match the lower bound on the communication complexity, but at the cost of performing more local computation (larger $K$). Nevertheless, we note that the $1/\epsilon^3$ lower bound in the i.i.d. setting is achieved by more complex algorithms than those considered in this paper. For example, Algorithm 2 in [9] utilizes updates in round $t$ that require evaluation of the gradient at the global model in round $t - 1$.

**Experimental Validation:** A challenge in numerically understanding the performance of FL algorithms with Markovian sampling is the strong impact of the mixing time of the data-generating process. In real data sets, such as the pollution data in Section 5, the mixing time cannot be easily controlled, which means that a comparison of the algorithms in different regimes is more challenging. To study the impact of the pseudo spectral gap or the mixing time, we instead relied on synthetic experiments in Section E.3.

