# OpenReview forum: "Streaming Federated Learning with Markovian Data"
_NeurIPS.cc/2025/Conference — NeurIPS 2025 poster_

### Official Review · Reviewer_hK1v · 2025-06-17

**Clarity:** 3
**Significance:** 3
**Originality:** 3
**Rating:** 4
**Confidence:** 3

**Summary:**

1. This paper establishes the upper bounds (of convergence rates) for Minibatch SGD and Local SGD (FL) on Markovian data.
2. This paper establishes the upper bounds of the momentum-based variants of Local SGD.

**Questions:**

**Questions**:

I am not familiar with Markov chain. So my questions are focused on the part related to Markov chain.

1. Section 4.1, Assumptions, Lines 214--215: What does $\tau$ mean?

2. Proof of Lemma B.1, Lines 581--582. What does $x_{t-1}^{K-1} = x$ mean? Can you elaborate on the change of measures?

3. Proof of Lemma B.1, Line 582. What does $|| \cdot ||_{\pi, \infty}$ mean? Can you elaborate on how to use Holder's inequality? You can write the form of Holder's inequality used here and specify how to use it.

4. Proof of Theorem B.3, Line 607. Can you elaborate on this equality?

5. Proof of Lemma B.5, Line 626, the first inequality. Is it possible to do as follows?

    $$
   \begin{align*}
   \xi_t &= \frac{1}{MK} \sum_{m,k} \eta^2  E_t \\| \sum_{j=0}^{k-1}\nabla f_m(w_t^{(m,j)};x_t^{(m,j)}) \\|^2 \\\\
   &\leq \frac{1}{MK} \sum_{m,k} \eta^2  \left( E_t \\| \sum_{j=0}^{k-1}\nabla f_m(w_t^{(m,j)};x_t^{(m,j)}) - \sum_{j=0}^{k-1}\nabla f_m(w_t;x_t^{(m,j)}) \\|^2 + E_t \\| \sum_{j=0}^{k-1}\nabla f_m(w_t;x_t^{(m,j)})- \sum_{j=0}^{k-1}\nabla F_m(w_t) \\|^2 + E_t \\|  \sum_{j=0}^{k-1}\nabla F_m(w_t) \\|^2 \right)
   \end{align*}
   $$

   The above inequality has three terms. Can the second term be bounded by [41, Theorem 3.7] (as done in Lemma B.1)? If so, does this cause a different bound for $\xi_t$? Does this cause different convergence rates for FL?

**Suggestions**:

1. Could you add the comparisons of upper bounds with Minibatch SGD and Local SGD with IID data for both without momentum and with momentum settings in a more straightforward way? One simple way is collect them in one table and highlight the main difference in different colors, just like the tables in [+6,+7]. This can help the readers see their differences more clearly.

   [+6] Minibatch vs Local SGD for Heterogeneous Distributed Learning, NeurIPS, 2020.

   [+7] Momentum Benefits Non-Iid Federated Learning Simply and Provably, ICLR, 2024.

2. Could you add the comparisons with SGD on Markovian data in a more straightforward way? One simple way is collect them in one table and highlight the main difference in different colors, just like the tables in [+6,+7].

3. Is it possible to remove the "streaming" in the title. Compared to the existing works focusing on FL with IID data [+5], this work study FL with Markovian data. Nevertheless, this work still considers the typical stochastic optimization, instead of online stochastic optimization. Thus, in my opinion, "streaming" may be unnecessary, and may misguide the readers into the works of online federated learning [+8,+9], which also claimed to address the problem of streaming data.

   I know that this may cause some revisions of the introduction. I think that [+10] (see their introduction) will be helpful.

   [+8] Asynchronous Online Federated Learning for Edge Devices with Non-IID Data, BigData, 2020.

   [+9] Online Federated Learning, CDC, 2021.

   [+10] On the Decentralized Stochastic Gradient Descent With Markov Chain Sampling, IEEE Transactions on Signal Processing, 2023.

**Scores**:

If character count is limited, please focus on Weaknesses.

1. I will raise my score to 4, as long as Weaknesses 1 and 2 are well explained. However, I cannot guarantee a higher score for now, as I am not an expert in Markov chains.
2. I have checked the proofs of Theorems 4.5--4.8 (without momentum) and have the questions above. Explaining them clearly will make this work more convincing and accessible.
3. The suggestions are not enforced. Nevertheless, I hope that the authors can consider them.

**Ethical Concerns:**

["NO or VERY MINOR ethics concerns only"]

**Final Justification:**

This paper establish the convergence guarantees of FedAvg (w/ or w/o momentum) on Markovian data.

The concern in my initial review is the non-vanishing term of the bounds (it is essentially the same concern of Reviewer T7vf). The responses of the authors seem reasonable for me, and they are also acknowledged by Reviewer T7vf.

Considering that I am not an expert of Markovian SGD, it is hard for me to verify their responses in detail, I decide to increase my rating to 4, and decrease my confidence to 3.

Lastly, I hope the authors will revise this paper as promised.

**Limitations:**

yes

**Quality:**

3

**Strengths And Weaknesses:**

**Strengths**:
1. The upper bounds of FL on Markovian data are novel (to the best of my knowledge) in the FL literature.
2. The momentum-based variants are also new for FL on Markovian data.
3. The amount of work in this paper is sufficient.
4. The paper is well-organized and well-written.

**Weaknesses**:

1. My **main concern** is that there exists one non-vanishing term (as $T$ approaches infinity) in the upper bounds (e.g., Theorems 4.5 and 4.7). This is quite different from the classical results of Markov chain SGD [+1,+2,+3,+4] and FL [+5], whose upper bounds approach zero as $T$ approaches infinity if an appropriate step size is chosen.

   [+1] Finite-Time Analysis of Markov Gradient Descent, TAC, 2022.

   [+2] Adapting to Mixing Time in Stochastic Optimization with Markovian Data, ICML, 2022.

   [+3] Stochastic Gradient Descent under Markovian Sampling Schemes, ICML, 2023.

   [+4] First Order Methods with Markovian Noise: from Acceleration to Variational Inequalities, NeurIPS, 2023.

   [+5] Tighter Theory for Local SGD on Identical and Heterogeneous Data, AISTATS, 2020.

2. Could you discuss the different measures used for Makrov chain in this paper and previous works? After checking some works of Markov chain SGD, I note that the mixing time is common (e.g., [+3]). So could you elaborate on why we use $C_{\infty}$ and $\nu_{ps}$, rather than the mixing time (or other measures). I am curious about this.

---

> ### Author Rebuttal · Authors · 2025-07-31
>
> We sincerely thank the Reviewer for their detailed and thoughtful feedback. We are pleased to see that the Reviewer values our contribution. Below, we address each of the Reviewer’s comments in turn.
>
> ### Main weaknesses
>
> > My main concern is that there exists one non-vanishing term (as $T$ approaches infinity) in the upper bounds (e.g., Theorems 4.5 and 4.7)...
>
> - We highlight that the non-vanishing term in Theorems 4.5, 4.7, 4.9 is scaled by $1/K$, where $K$ is the batch size (or the number of local steps in local SGD). As a consequence, $K$ can be chosen to be a function of $T$ such that the term vanishes. For example, choosing $K = T$ in Theorem 4.5 for Minibatch SGD or $K = \sqrt{T}$ and $\eta = 1/T$ in Theorem 4.7 for Local SGD, all terms in the bounds converge to zero as $T \rightarrow \infty$. For example, with minibatch SGD we have the following modification of Theorem 4.5.
>
> - *Theorem: Suppose the conditions of Theorem 4.5 hold, and $K = T$. Then,*
> $$
> \frac{1}{T} \sum_{t=0}^{T-1} E [ || \nabla F(w_t) ||^2 ] \leq \mathcal{O} \left(\frac{\Delta_0}{\gamma T} + \frac{C_{\infty}\sigma^2}{\nu_{ps}MT} \right).
> $$
>
> - Note that letting $K$ scale with $T$ does not lead to any improvement in the sample complexity over our corollaries 4.6,4.8,4.10. Hence, we presented our results with $K$ independent of $T$ as, in practice, $K$ may be determined by the arrival rate of the data samples and the buffer size of each client.
>
> - We highlight that scaling the number of samples utilized in each iteration is also the method of [3,13] (for the centralized Markov SGD) to ensure that all terms vanish as the number of iterations diverges.
>
> - There is a theoretical justification for why the variance term in our bounds cannot be directly controlled by the step size. By a standard decomposition from the $L$-smoothness condition:
> $$
> E_t [ F(w_{t+1}) ] \leq F(w_t) - \gamma/2 (1-2\gamma L) || \nabla F(w_t) ||^2 + \gamma/2 || E_t [g_t] - \nabla F(w_t) ||^2 +\gamma^2 L E_t [ || g_t - \nabla F(w_t) ||^2 ]
> $$
> where $b_t = || E_t [g_t] - \nabla F(w_t) ||^2$ and $v_t = E_t [ || g_t - \nabla F(w_t) ||^2$ denotes the bias and the variance of the gradient estimate, and $\mathbb{E}_t$ denotes the expectation conditioned on the history up to time t. This bias arises from the Markovian structure of the data, and does not appear in the i.i.d. setting. Observe that the bias is scaled by the step size $\gamma$. Since the term $||\nabla F(w_t) ||^2$ is also scaled by $\gamma$, the bias cannot be directly controlled by the choice of step size. By Jensen's inequality, we have $b_t \leq v_t$, hence the bias also contributes to the "variance" term arising in our main results. This leads to a non-vanishing "variance" term in our results as the step size decreases.
>
> - We now give additional details of the techniques used in [3, 12, 13, 15] to control this bias, and obtain a vanishing variance term. In short, they introduce other hyper-parameters that can be tuned to make the bias vanish, instead of tuning directly the step size.
>     - [12, 15] decompose the bias term into other terms containing delayed gradient. This delay is later chosen to be scaled with $1/\sqrt{T}$. This technique requires a constraint on the minimum number of iterations (analogous to communication rounds $T$) at the order of $\tilde{\mathcal{O}} (\tau_{\mathrm{mix}})$. As revealed in [Table 1, 3], [12] also contains hidden factors that can be exponential in $\tau_{\mathrm{mix}}$. The bound for non-convex function in [12] (Theorem 3 in the arxiv version) also requires a projection onto the unknown set that contains all the optima.
>     - [3, 13], as discussed above, use minibatch gradient estimates. Precisely, at each iteration, a random batch size is drawn from a truncated geometric distribution. Hence, a new hyper-parameter is introduced which is the truncation threshold $m$. [3, 12] shows that $b_t$ scales inversely with $m$, which is then chosen to be scaled with $1/\sqrt{T}$.
>
> - We highlight that the sample complexities obtained in our paper are of the order of $1/(M\epsilon^4)$, which is the same as those achieved in [3, 12, 13, 15] scaled inversely with the number of clients $M$. However, our assumptions on the underlying Markov processes are weaker (see Section 3.2 in our paper). As not every convergence result in [3, 12, 13, 15] are stated as sample complexity, we refer to [Table 1, 3] for a detailed comparison in term of sample complexity.
>
> > Could you discuss the different measures used for Makrov chain in this paper and previous works?...
>
> - The use of $C_{\infty}$ and $\nu_{ps}$ comes from the fact that we do not impose uniform ergodicity assumption on the underlying Markov chains as in [3, 12, 15]. This suggests that our results can be applied to a broader class of Markov chains. [13] does not require uniform ergodicity, but their bounds rely strongly on the bounded gradient norm assumption.
>
> - More precisely, the pseudo spectral gap $\nu_{ps}$ arising in our bounds is due to a variance bound for Markov chains [Theorem 3.7, 41], which applies for very general Markov chains (non-reversible and non-uniformly ergodic). However, this variance bound only holds for stationary chains (chains start from the stationary distribution). The term $C_{\infty}$ appears in a change of measure to account for this non-stationarity.
>
> - In contrast, [3] relies on uniform ergodicity to deal with non-stationarity, but this can yield a dependence on the square of the mixing time [Section B.1, 3], and applying blindly their technique in the FL setting does not show the linear speedup. Recent works on FL with Markovian data [35] with linear speedup only consider stationary and uniformly ergodic Markov processes.
>
> - We provide a characterization of $C_{\infty}$ and $\nu_{ps}$ under more restricted settings in Appendix B.6 of our paper, showing that we can recover the dependence on the mixing time as in [3, 12, 13, 15].
>
> ### Questions
>
> > Section 4.1, Assumptions, Lines 214--215: What does $\tau$ mean?
>
> - This is a typo. The problem classes defined at the end of Section 4.1 only depend on $C_{\infty}$ and $\nu_{ps}$. We will fix this in the revised version of the paper.
>
> > Proof of Lemma B.1, Lines 581-582...
>
> - Here $x_{t-1}^{K-1}$ denotes all the Markov samples used by the clients in the last local step of round $t-1$. In the proof of Lemma B.1, we first need to take the conditional expectation with respect to the history up to round $t$, so that the randomness now depends only on the Markov samples arriving in round $t$. Due to the memoryless property of Markov chain [37, Proposition 3.4.3], conditioning on the past is equivalent to conditioning only on the most recent sample. We write $x_{t-1}^{K-1}=x$ to highlight that we are conditioning on $x_{t-1}^{K-1}$. Denote by $q$ the probability measure $P ( {x_{t-1}^{K-1}}=x, .)$. The conditional expectation $E_t[.]$ is equivalent to $E_q[.]$. The Markov samples used in round $t$, conditioned on $x_{t-1}^{K-1}$, is a Markov chain with starting distribution $q$. The change of measure comes in to make this Markov chain stationary (i.e. start from the stationary measure $\pi$), so that [Theorem 3.7, 41] can be applied.
>
> > Proof of Lemma B.1, Line 582...
>
> - $\left\lVert \cdot \right\rVert_{\pi, \infty}$ denotes the norm in the $L^{\infty} (X, \pi)$ space of measurable function $f: X \rightarrow \mathbb{R}$. More precisely,
> $$
> \left\lVert f \right\rVert_{\pi, \infty} = \inf \left\\{ C \geq 0, | f(x) | \leq C~ \text{for}~ \pi \text{-almost everywhere}~ x \right\\}
> $$
>
> - This is an equivalent way to write the essential supremum in Assumption 3.2 of our paper. We write this way to make it clearer that we use Holder's inequality in this step. The exact form of Holder's inequality used here is:
> $$
> \mathbb{E} [|XY|] \leq \mathbb{E} [|X|^p] ^ {1/p} \mathbb{E} [|Y|^q]^{1/q},
> $$
> for two random variables $X, Y$, and $1/q+1/p=1$. We then apply this inequality with $p=\infty, q=1$, $X$ is the Radon-Nikodym derivative, and $Y$ is the term with the square Euclidean norm.
>
> > Proof of Theorem B.3...
>
> - This equality follows from the fact that $\hat{w_T}$ is sampled uniformly at random from the iterates $w_0, \dots, w_{T-1}$, independently from the randomness during training.
>
> > Proof of Lemma B.5...
>
> - Thank you for your suggestion. Indeed, what you suggest is totally possible, but doing this will make the term $1/\nu_{ps}$ appear in the drift bound, and later on, appear in the communication complexity. We note that $1/\nu_{ps}$ can be upper-bounded by the mixing time. Thus, this technique will result in a worse communication complexity than the one achieved in Theorem 4.7 of our paper.
>
> ### Suggestion
>
> - We greatly appreciate the Reviewer’s thoughtful suggestions and will incorporate all of them into the revised version of our paper.
>
> #### References
> [3] Aleksandr Beznosikov, Sergey Samsonov, Marina Sheshukova, Alexander Gasnikov, Alexey Naumov, and Eric Moulines. First order methods with markovian noise: from acceleration to variational inequalities. Neurips 2023.
>
> [12] Thinh T. Doan, Lam M. Nguyen, Nhan H. Pham, and Justin Romberg. Finite-Time Analysis of Stochastic Gradient Descent under Markov Randomness. arXiv:2003.10973.
>
> [13] Ron Dorfman and Kfir Yehuda Levy. Adapting to mixing time in stochastic optimization with Markovian data. ICML 2022.
>
> [15] Mathieu Even. Stochastic gradient descent under Markovian sampling schemes. ICML 2023.
>
> [35] Paul Mangold, Sergey Samsonov, Safwan Labbi, Ilya Levin, Reda ALAMI, Alexey Naumov, and Eric Moulines. SCAFFLSA: Taming heterogeneity in federated linear stochastic approximation and TD learning. Neurips 2024.
>
> [37] Meyn S, Tweedie RL, Glynn PW. Markov Chains and Stochastic Stability. 2nd ed. Cambridge University Press; 2009.

---

> ### Comment · Reviewer_hK1v · 2025-08-05
> **Further suggestions**
>
> Thanks for the rebuttal. The justification for the non-vanishing term (also noted by Reviewer T7vf) sounds reasonable for me. So I will adjust my rating accordingly, after the discussion with other reviewers and AC.
>
> Suggestions:
>
> 1. I hope the authors discuss the difference between this work and the existing works that focusing on Markovian data, especially the limitations of the existing works. A more thorough discussion, beyond what is included in the current rebuttal, could be provided in the revised version. In particular, the authors should explain why they choose to use the current approach instead of the existing approaches developed by the existing works.
> 2. A more careful check for the proof is still needed to avoid the possible typos.
> 3. The responses to the other comments can also be added in the revised version to improve the readability.

---

> > ### Author Response · Authors · 2025-08-05
> > **Response to Reviewer's comment**
> >
> > We sincerely thank the Reviewer for re-evaluating our submission in light of our rebuttal. We greatly appreciate the time and effort you dedicated to carefully considering our response. We are glad to hear that your concern regarding the non-vanishing variance term in our results has been addressed, and we hope that our rebuttal has also resolved your remaining questions. Please let us know if any further clarifications are needed. We will certainly incorporate your suggestions, along with those from the other reviews, into the revised version of our paper.

---

> > ### Author Response · Authors · 2025-08-06
> > **Response to Reviewer's comment**
> >
> > > I hope the authors discuss the difference between this work and the existing works that focusing on Markovian data, especially the limitations of the existing works. A more thorough discussion, beyond what is included in the current rebuttal, could be provided in the revised version. In particular, the authors should explain why they choose to use the current approach instead of the existing approaches developed by the existing works.
> >
> > - We would like to further clarify the motivation of approach and how it differs from existing works on Federated Learning (FL) with Markovian data.
> >
> > - First, a key objective in FL is to reduce the communication overhead by leveraging local computation. Our approach directly aligns with this goal. Specifically, we show that increasing local computation enables FL algorithms with Markovian sampling to match the communication complexity lower bound established for the i.i.d. setting. This demonstrates that the benefits of local updates persist even in non-i.i.d. (Markovian) regimes. Our result contrasts with the approach in [12, 15] (established for centralized or token-passing scenarios), which requires a constraint on the minimum number of iterations (analogous to the communication cost $T$). Moreover, we treat the number of local steps $K$ as an independent parameter, which can be influenced by practical constraints such as the data arrival rate or the buffer size of client devices. This modeling choice, in our view, better reflects real-world FL scenarios than the approaches in [3,13] that tightly couple $K$ to the step size or the number of communication rounds $T$.
> >
> > - Second, regarding the treatment of Markovian data, our analysis introduces alternative measures, such as $C_{\infty}$ and $\nu_{ps}$, to quantify the effect of temporal correlations. These measures differ from the commonly used mixing time in prior works, which relies on uniform ergodicity and stationarity assumptions, and arise from our use of weaker assumptions on the underling Markov processes. As the standard assumptions of uniform ergodicity and stationarity are already difficult to verify, we believe that considering weaker assumptions makes our results more broadly applicable, and bring the theoretical analysis one step closer to practical deployment scenarios.

---

### Official Review · Reviewer_T7vf · 2025-07-01

**Clarity:** 2
**Significance:** 2
**Originality:** 3
**Rating:** 4
**Confidence:** 3

**Summary:**

This paper addresses the challenges of Federated Learning (FL) in the context of streaming, non-i.i.d. data, specifically focusing on Markovian data streams, which frequently arise in real-world applications such as health and environmental monitoring. Unlike standard FL setups where clients train on static datasets, the authors consider data generated by time-homogeneous Markov processes, which introduces temporal dependencies and non-stationarity. The authors conduct a rigorous theoretical analysis of three FL algorithms in this setting: (i) Minibatch SGD; (ii) Local SGD, and (iii) Local SGD with Momentum (Local SGD-M). In addition to the theoretical contributions, the paper includes experiments on a regression task using a real-world dataset to illustrate the behavior of the algorithms under Markovian sampling.

**Questions:**

1) Can you clarify why the variance term in your convergence bounds is not controlled by the step-size, unlike in [1]? Can you either refine the analysis or show empirically that this term does not vanish with a smaller step-size?

2) Would it be possible to rerun or supplement the experiments with step-size schedules that follow the theoretical scaling with respect to $K$ or a grid search?

**Ethical Concerns:**

["NO or VERY MINOR ethics concerns only"]

**Final Justification:**

The authors have adequately addressed my concerns, especially regarding the non-compressible 'variance' term. Their response clarified the key points I raised, and I find the rebuttal satisfactory. I am therefore increasing my score.

**Limitations:**

yes

**Paper Formatting Concerns:**

no concern

**Quality:**

3

**Strengths And Weaknesses:**

**Strengths**

1) The paper tackles a realistic and underexplored setting in FL by modeling client data streams as non-stationary Markov processes, a framework that reflects the dynamics of many physical and biological systems.

2) The theoretical analysis is rigorous and well-written, and characterizes the convergence behavior of Minibatch SGD and Local SGD under Markovian sampling. In particular, the results confirm that linear speedup (i.e., sample complexity scaling with $1/M$) is achievable under standard smoothness and bounded heterogeneity assumptions.

3) An important contribution is the analysis of Local SGD with Momentum, which is shown to mitigate client drift due to data heterogeneity under Markovian streaming, achieving convergence guarantees without the need for gradient dissimilarity assumptions.

**Weaknesses**

1) *Main concern* : The variance term in the convergence bounds is independent of the step-size, which seems suboptimal. Intuitively, one would expect the variance term to decay as the step size decreases since smaller steps imply smaller parameter movement from initialization. This contradicts both empirical intuition and existing results in the literature (e.g., [1]), where the variance is weighted by the step-size. The authors should either provide theoretical justification for why such an "incompressible" variance term appears, or offer empirical evidence of its presence and persistence across different step-size regimes.

2) In Section 4.3 ("Impact of Markovian Data"), the paper references [2] and asserts that the bias term in Theorem 2 of [2] is persistent and uncontrollable even with increased local computation. However, the bias in [2] originates from client heterogeneity, not from Markovian sampling. This remark is misleading and should be relocated to the section discussing heterogeneity.

3)  In the experimental section, the choice of hyperparameters lacks consistency with the theoretical findings. For example, the theory suggests that the step size should scale as $\mathcal{O}(K)$, yet the experiments use a fixed step size across different values of $K$. This makes it difficult to isolate whether stagnation in performance is due to the variance term, the heterogeneity term, or simply hyperparameter mismatch. I strongly suggest including either experiments where step-size is scaled according to theory, or to select the hyperparameters according to a grid search, to better support the empirical conclusions drawn in the paper.

[1] Zhu,F. ,Mitra,A., and Heath, R.W. (2025) . Achieving tighter finite-time rates for heterogeneous federated stochastic approximation under markovian sampling.

[2] Han Wang, Aritra Mitra, Hamed Hassani, George J. Pappas, and James Anderson. Federated TD learning with linear function approximation under environmental heterogeneity. Transactions on Machine Learning Research, 2024

---

> ### Author Rebuttal · Authors · 2025-07-31
>
> We are happy that the Reviewer appreciates the contribution of our work! We address the Reviewer's comments below.
>
> > Main concern: The variance term in the convergence bounds is independent of the step-size, which seems suboptimal...
>
> > Can you clarify why the variance term in your convergence bounds is not controlled by the step-size, unlike in [1]? Can you either refine the analysis or show empirically that this term does not vanish with a smaller step-size?
>
> - The main concern of the reviewer is the presence of a variance term (dependent on the variance of the gradient estimator) in Theorems 4.5,4.7,4.9, which is not scaled by the step size. We first highlight that our results apply for non-convex objectives with Markovian sampling in the federated setting, while existing results motivated by reinforcement learning (e.g., TD learning) with linear function approximation are in the (strongly) convex setting (see lines 40-42), **including [Zhu,F. ,Mitra,A., and Heath, R.W. (2025)] pointed by the reviewer**. We were not aware of this work which appeared online one month before NeurIPS and we thank the reviewer for pointing it out. In the revised version of the paper we will also include this reference.
> - We first give a theoretical justification that clarifies why adjusting the step size alone offers limited leverage for controlling the variance term. By a standard decomposition from the $L$-smoothness condition:
> $$
> E_t [ F(w_{t+1}) ] \leq F(w_t) - \gamma/2 (1-2\gamma L) || \nabla F(w_t) ||^2 + \gamma/2 || E_t [g_t] - \nabla F(w_t) ||^2 +\gamma^2 L E_t [ || g_t - \nabla F(w_t) ||^2 ]
> $$
> where $b_t = || E_t [g_t] - \nabla F(w_t) ||^2$ and $v_t = E_t [ || g_t - \nabla F(w_t) ||^2$ denotes the bias and the variance of the gradient estimate, and $\mathbb{E}_t$ denotes the expectation conditioned on the history up to time t.  Observe that the bias is scaled by the step size $\gamma$. Since the term $||\nabla F(w_t) ||^2$ is also scaled by $\gamma$, the bias cannot be directly controlled by the choice of step size without sacrificing the reduction of the objective. Note that by Jensen's inequality, $b_t \leq v_t$, hence the bias also contributes to the "variance" term arising in our main results. This lead to a non-vanishing "variance" term in our results as the step size decreases.
>
> - For smooth non-convex objectives, we are aware of only three papers [3,13,15] that have a variance term vanishing as the number of iterations diverges. Note that the three papers consider centralized or token-passing settings as opposed to the federated setting in our paper. These papers make the variance term vanish by introducing auxiliary parameters that can be tuned to scale the bias with the step size or with $1/\sqrt{T}$. This is achieved:
>     - in [15] by decomposing the bias into other terms that contain delayed gradients, and scaling such delay as $1/\sqrt{T}$. This technique requires a constraint on the minimum number of iterations $T$ (analogous to communication rounds $T$).
>     - in [3,13] by using a minibatch gradient estimate at each iteration (instead of using just one sample). The minibatch size utilized is then chosen to be scaled inversely with the step size [13], or with $\sqrt{T}$ [3].
>
> - We highlight that the sample complexities obtained in our paper is at the order of $1/(M \epsilon^4)$, which is the same as those achieved in [3, 13, 15] and scale inversely with the number of clients $M$. However, our assumptions on the underlying Markov processes are weaker (see Section 3.2 in our paper).
> - In our paper, we considered $K$ as independent of both the step size and the number of rounds $T$, because we think that in real deployments the value of $K$ is dictated by the data-arrival rate and client's buffer capacity. Nonetheless, following the approach of [3, 13], $K$ grow with $T$ or inversely with the step size, which would indeed drive the variance term to zero as $T \rightarrow \infty$. Note, however, that this would not lead to any improvement in sample complexity.
>
> > In Section 4.3 ("Impact of Markovian Data"), the paper references [2] and asserts that the bias term in Theorem 2 of [2] is persistent and uncontrollable even with increased local computation...
>
> - We thank the Reviewer for their comment and after re-reading [2], we agree that this reference would be better suited in the section discussing heterogeneity. In the revised version of the paper, we will make this modification.
>
> > In the experimental section, the choice of hyperparameters lacks consistency with the theoretical findings...
>
> > Would it be possible to rerun or supplement the experiments with step-size schedules that follow the theoretical scaling with respect to $K$ or a grid search?
>
> - As suggested, we have performed additional experiments to investigate the scaling of the step size. As the reviewer notes, for Local SGD, the choice of the local step size can impact performance. For moderate values of the initial suboptimality gap and heterogeneity:
>     - If $K$ is large, then the bound is dominated by the heterogeneity term.
>     - If $K$ is small, then the bound is dominated by the variance term.
> - To empirically validate this, we modified the setup in Section 5 to scale the local learning rate as $\eta = b / K$, with two base values: $b=0.1$ and $b=0.01$. This replaces the fixed learning rate $\eta = 0.01$ used in the original experiments. We report below the mean gradient norm after 80 communication rounds, along with the 95% confidence interval over 5 random seeds:
>
>     | $\eta$   | $K=10$              | $K=50$              | $K=100$             |
>     | -------- | ------------------- | ------------------- | ------------------- |
>     | $0.1/K$  | $0.00506 ± 0.00076$ | $0.00307 ± 0.00041$ | $0.00233 ± 0.00030$ |
>     | $0.01/K$ | $0.01248 ± 0.00279$ | $0.00921 ± 0.00210$ | $0.01190 ± 0.00245$ |
>
> - We report here the bound for Local SGD from Theorem 4.7:
> $$
> \frac{1}{T} \sum_{t=0}^{T-1} E [ || \nabla F(w_t) ||^2 ] \leq \mathcal{O} \left( \frac{\Delta_0}{\eta K T} + \frac{C_{\infty} \sigma^2}{\nu_{ps}MK} + \frac{L K \eta (\theta^2+\sigma^2)}{\delta^2}  \right)
> $$
> - Choosing the step size as $\eta \propto 1/K$ leaves the first and third error terms unchanged with  $K$, while the second term decays with increasing $K$. Hence, when this second term dominates for small $K$, the gradient norm $||\nabla F(w)||^2$ should decrease before flattening out at a positive plateau. The patterns shown in each row align with this qualitative prediction.
>
> - The comparison of the two rows support the fact that the variance term cannot be controlled by the step size. Indeed, for a fixed $K$, reducing the step size by a factor of $10$ does not improve the performance. On the contrary, the gradient norm appears to increase for smaller $\eta$. This can be explained by the first term in the bound, associated with the suboptimality gap, which scales inversely with the step size.
>
> #### References:
>
> [3] Aleksandr Beznosikov, Sergey Samsonov, Marina Sheshukova, Alexander Gasnikov, Alexey Naumov, and Eric Moulines. First order methods with markovian noise: from acceleration to variational inequalities. Neurips 2023.
>
> [13] Ron Dorfman and Kfir Yehuda Levy. Adapting to mixing time in stochastic optimization with Markovian data. ICML 2022.
>
> [15] Mathieu Even. Stochastic gradient descent under Markovian sampling schemes. ICML 2023.
>
> [Zhu,F. ,Mitra,A., and Heath, R.W. (2025)] Zhu,F. ,Mitra,A., and Heath, R.W. (2025) . Achieving tighter finite-time rates for heterogeneous federated stochastic approximation under markovian sampling.

---

> > ### Comment · Reviewer_T7vf · 2025-08-04
> >
> > I appreciate the authors’ detailed and informative response. My concerns have been addressed, and I will accordingly revise my score.

---

> > > ### Author Response · Authors · 2025-08-05
> > > **Response to reviewer's comment**
> > >
> > > We sincerely thank the Reviewer for re-evaluating our submission in light of our rebuttal. We greatly appreciate the time you dedicated to carefully considering our response, and we’re glad to hear that your concerns have been addressed.

---

### Official Review · Reviewer_CkzJ · 2025-07-02

**Clarity:** 3
**Significance:** 4
**Originality:** 3
**Rating:** 5
**Confidence:** 3

**Summary:**

The work provides methods for federated learning in the setting where clients hold a sliding window of K most-recent samples that evolve as a time-homogeneous Markov chain converging to a stationary distribution.

**Questions:**

1.) Would it be preferred to have a small $\theta$ and $\delta$, in the context of the  bounded gradient dissimilarity (BGD) assumption? If so, how would you ensure that in your work?

2.) How would extending to mini-batch updates (multiple correlated samples) change the convergence results?

3.) What is local local SGD on line 212 and how is it different from local SGD? Is local local SGD a typo or refers to Local SGD with Momentum?

4.) What is the difference in the setup in proposition C.10 Vs. the equation between line 740 and 741 that give mixing time results?

**Ethical Concerns:**

["NO or VERY MINOR ethics concerns only"]

**Final Justification:**

The authors have done a thorough job and clarified all the main concerns. Based on that and rest of the discussions, I uphold my score at 5.

**Limitations:**

These have been described in great detail.

**Quality:**

3

**Strengths And Weaknesses:**

Strengths:
The convergence bounds provided are sharp. The method has a good communication efficiency.
The problem is well motivated as such temporal streams of data are quite common in the federated setting.

It provides some mixing-time results to understand the convergence to a statioary target distribution.

Main weakness:
1.) In order to guarantee convergence, they suggest that many local steps are needed. Can you share some more info about this?

---

> ### Author Rebuttal · Authors · 2025-07-31
>
> We really appreciate the Reviewer's positive comments about our paper! We address the Reviewer's concerns below.
>
> ### Main weaknesses
>
> > In order to guarantee convergence, they suggest that many local steps are needed. Can you share some more info about this?
>
> - It is true that in order to guarantee convergence, there is a constraint on the number of local steps. The reason for this is that the term $\frac{C_{\infty}\sigma^2}{\nu_{ps}MK}$ in our bounds corresponds to the bias of the gradient estimates and cannot be controlled by the step size. A similar issue is already present even for centralized Markov SGD, where the bias is often controlled by increasing the batch size (see, e.g [3]). Other works [12, 15], also in the centralized setting, instead require a large number of iterations to control the Markovian bias—an approach that would translate to higher communication cost in federated learning. In contrast, our analysis is more aligned with the core aim of FL---trading extra local computation for fewer communication rounds.
>
> ### Questions
>
> > Would it be preferred to have a small $\theta$ and $\delta$, in the context of the bounded gradient dissimilarity (BGD) assumption? If so, how would you ensure that in your work?
>
> - Ideally, in the homogeneous setting (all clients learning task are the same), $\theta = 0$ and $\delta = 1$ would minimize the heterogeneity term for local SGD in Theorem 4.7. On the other hand, the heterogeneity parameters $\theta$ and $\delta$ are determined by the gradient functions $\nabla F_m, m \in [M]$, which depends on the data distribution $\pi_m$ and the local objective $f_m$. As a consequence, $\theta$ and $\delta$ can only be changed by modifying the data distribution or the learning task. For example local data distributions may be made closer if clients are allowed to share at least a part of their dataset, something that is not usually envisaged in federated learning, but it is an option in the so-called fog learning (e.g. [Hosseinalipour et al. 2020])
>
> > How would extending to mini-batch updates (multiple correlated samples) change the convergence results?
>
> - Thank you for your great question! Indeed our analysis can be generalized to consider minibatch updates. Precisely, the number of samples utilized per communication round appears in the term $E_t [ || g_t - \nabla F(w_t) ||2 ]$, where $E_t$ is the expectation conditioned on information available up to time $t$ and $g_t = \frac{1}{MK}\sum_{m,k} \nabla f_m(w_t^{(m,k)};x_t^{(m,k)})$. As a consequence, if mini-batch updates are utilized with the same number of samples $K$ utilized per communication round, then the variance term $\frac{C_{\infty}\sigma^2}{\nu_{ps}MK}$ will not change. However, in local SGD, if fewer local steps are taken, this can reduce the impact of heterogeneity (corresponding to the last term in Theorem 4.7). In the extreme case where only one local step is used (with a mini-batch of $K$ samples), this corresponds to mini-batch SGD, for which the convergence bound is given in Theorem 4.5. In the revised version of our paper, we will add discussions about this extension.
>
> > What is local local SGD on line 212 and how is it different from local SGD? Is local local SGD a typo or refers to Local SGD with Momentum?
>
> - Many thanks for your careful reading. The “local local SGD” reference was a typo and will be replaced with local SGD with Momentum in a revised version.
>
> > What is the difference in the setup in proposition C.10 Vs. the equation between line 740 and 741 that give mixing time results?
>
> - Proposition C.10 provides a general characterization of the mixing time of the system Markov chain in terms of the mixing times of the individual mixing times of the client chains. In the case the system Markov chain is also uniformly ergodic (as discussed in Appendix B.5), then the pseudo spectral gap $\nu_{ps}$ of the system Markov chain can be bounded in terms of its mixing time (see lines 733-734). The result of Proposition C.10 can then be applied (see lines 735-736), which leads to the equation you indicate between lines 740 and 741.
>
>
> #### References
>
> [3] Aleksandr Beznosikov, Sergey Samsonov, Marina Sheshukova, Alexander Gasnikov, Alexey Naumov, and Eric Moulines. First order methods with markovian noise: from acceleration to variational inequalities. Neurips 2023.
>
> [12] Thinh T. Doan, Lam M. Nguyen, Nhan H. Pham, and Justin Romberg. Finite-Time Analysis of Stochastic Gradient Descent under Markov Randomness. arXiv:2003.10973.
>
> [15] Mathieu Even. Stochastic gradient descent under Markovian sampling schemes. ICML 2023.
>
> [Hosseinalipour et al. 2020] S. Hosseinalipour, C. G. Brinton, V. Aggarwal, H. Dai and M. Chiang, From Federated to Fog Learning: Distributed Machine Learning over Heterogeneous Wireless Networks, IEEE Communications Magazine 2020.

---

> ### Comment · Area_Chair_RZAi · 2025-08-05
>
> Dear Reviewer,
>
> To facilitate further evaluation of the submission, please take a moment to respond to the authors’ rebuttal.
>
> Best,
>
> AC

---

> ### Comment · Reviewer_CkzJ · 2025-08-05
> **Thanks for responses. Additional questions.**
>
> Thanks to the authors for their detailed rebuttal. Looking more carefuly at your proofs, I have some more questions.
>
> 1.) In the proof of Lemma 4.1 in the convergence analysis where you bound \mathbb{E}_\pi\left[\left\|\nabla f_m-\nabla F_m\right\|^2\right] \leq \frac{4 \sigma^2}{M K \nu_{p s}}, you use 41, Theorem 3.7 that applies to scalar-valued functions, but $\nabla f_m$ ​ is vector-valued. The proof sums over the dimensions instead, but the theorem’s concentration inequality may not hold for vectors without additional justification, especially for the markov setting. You may need some other concentration inequality.
> 2.) Can you share more insights on the applicability of Lemma B.8 to your update which has a non-standard form of the kind v_{t+1}=\frac{1}{\eta M K} \sum_m\left(w_t-w_t^{(m, K)}\right)?

---

> > ### Author Response · Authors · 2025-08-06
> > **Response to Reviewer's comment**
> >
> > We first thank the Reviewer for acknowledging our rebuttal. Below we answer the Reviewer follow-up questions in turn.
> >
> > > 1.) In the proof of Lemma 4.1...
> >
> > - We believe that the Reviewer is referring to **Lemma B.1** in our paper. Indeed, [Theorem 3.7, 41] applies only to scalar-valued functions, while $\nabla f_m$ in the proof of Lemma B.1 are vector-valued functions. To address this, as detailed in lines 584-585, we write the square norm of the vector-valued function as the sum of the square of its components, and then apply [Theorem 3.7, 41] to each of these component.
> >
> > > 2.) Can you share more insight on the applicability of Lemma B.8 to your update...
> >
> > - Lemma B.8 serves as a descent lemma for the convergence proof of Local SGD with Momentum (Local SGD-M). Its proof follows the same steps  as in the proof of Lemma B.2, which establishes a descent lemma for Minibatch SGD. In both case, the proof begins with the $L$-smoothness condition (see lines 596-597). The key difference lies in the update rule: for Local SGD-M, the global update at round $t$ is given by:
> > $$
> > w_{t+1} = w_t - \gamma v_{t+1}
> > $$
> > where $v_{t+1}$ is the **average of all local updates** performed by the clients during round $t$.
> >
> > - The term $\Xi_t = E[ || \nabla F(w_t) - v_{t+1} ||^2]$ in Lemma B.8 is the deviation between the global update used by the server at round $t$ and the true gradient of the global loss function $F$. This is directly analogous to the term $E_t [|| \bar{g_t} - \nabla F(w_t) ||^2]$ in Lemma B.2 for Minibatch SGD (or $E_t [|| g_t - \nabla F(w_t) ||^2]$ in Lemma B.5 for Local SGD). In those cases, $\bar{g_t}$ (or $g_t$) is the **average of all gradients** computed by all clients in round $t$, and hence the difference between the global update and the true gradient can be bounded by applying directly Lemma B.1 (possibly with some additional drift terms in the case of Local SGD).
> >
> > - For Local SGD-M, the global update $v_{t+1}$ can be rewritten as:
> > $$
> > v_{t+1} = \frac{\beta}{MK} \sum_{m=1}^M \sum_{k=0}^{K-1} + (1-\beta) v_t,
> > $$
> > where the present of the momentum term $(1-\beta)v_t$ carries information about the updates performed in the past. This allows us to build a recursive bound for $\Xi_t$ as in Lemma B.9 to Lemma B.12, without relying on any heterogeneity assumption to bound the drift terms caused by local updates as for Local SGD.

---

> ### Comment · Reviewer_CkzJ · 2025-08-08
> **Clarifications**
>
> There maybe a better concentration bound for the vector form as a whole that is better applicable as opposed to the component-wise bound, as [Theorem 3.7, 41] applied component-wise might be a looser result.
>
> Thanks for sharing the detailed insights on the applicability of Lemma B.8. This suffices, and it might be of value to include this as a proof sketch or addendum in the appendix. This enhances the readability of your lengthy draft. I keep my score.

---

> ### Author Response · Authors · 2025-08-08
>
> Many thanks for your careful reading. We would like to highlight that using [Theorem 3.7, 41] component-wise does not induce any additional factor dependent on the dimension of the data in our final bound. We will provide further insights into the proof in the revised version of our paper.

---

### Official Review · Reviewer_iDzF · 2025-07-03

**Clarity:** 3
**Significance:** 3
**Originality:** 3
**Rating:** 4
**Confidence:** 3

**Summary:**

The paper presents a rigorous theoretical and empirical study of federated learning (FL) under Markovian streaming data, a realistic and understudied scenario in FL. The key contribution lies in providing convergence guarantees and communication/sample complexity bounds for Minibatch SGD, Local SGD, and Local SGD with Momentum (Local SGD-M), all under non-i.i.d., temporally correlated data generated by Markov chains. The work is well-motivated, theoretically sound, and empirically validated.

**Questions:**

1. The paper evaluates performance on a single real-world dataset (air-quality time series). How generalizable are the findings across domains?
2. You suggest that momentum-based methods (SGD-M) mitigate heterogeneity without requiring control variates (e.g., SCAFFOLD). Could you provide direct empirical comparison with control variate-based methods, such as SCAFFOLD or SCAFFLSA, under the Markovian setting?
3. The theoretical results require careful tuning of parameters like step sizes $\gamma$, $\eta$, and momentum $\beta$ based on unknown quantities. Is there an adaptive strategy or heuristic the authors can recommend for real-world deployment?
4. All your evaluation metrics focus on gradient norm as the convergence criterion. Can you report test accuracy or loss (e.g., RMSE on PM2.5) to give a sense of practical performance?

**Ethical Concerns:**

["NO or VERY MINOR ethics concerns only"]

**Final Justification:**

4

**Limitations:**

yes

**Quality:**

3

**Strengths And Weaknesses:**

## Strengths
- The paper is well-organized, beginning with a strong motivation and providing precise mathematical modeling.
- The paper presents a rigorous theoretical analysis of FL under Markovian data streams, an underexplored yet realistic setting.
- It provides tight convergence bounds for Minibatch SGD, Local SGD, and Local SGD-M, under well-stated assumptions.
- Detailed proofs and assumptions are provided in the appendix; results are reproducible with shared code and setup.
- Empirical validation on real-world, temporally dependent data strongly supports the theoretical findings.

## Weaknesses
- While the theoretical results are strong, some assumptions (e.g., bounded Radon-Nikodym derivative, fixed spectral gap) may not be easily verifiable in practice.
- The experimental section is somewhat limited, only one real dataset is used, and synthetic settings are only briefly mentioned.
- The practical implications for deployment in real FL systems are not deeply explored. For example, how to estimate spectral gap or choose $K$ in practice is left unaddressed.

---

> ### Author Rebuttal · Authors · 2025-07-31
>
> We are happy that the Reviewer appreciated our solid theoretical results! We address the Reviewer's specific comments point-by-point below.
>
> ### Weaknesses
>
> > While the theoretical results are strong, some assumptions (e.g., bounded Radon-Nikodym derivative, fixed spectral gap) may not be easily verifiable in practice.
>
> - We highlight that our assumptions on the underlying Markov chain are weaker than those often used in the literature of SGD, which allows for more general data streams' models. For example, we do not impose the condition of uniform ergodicity, and rather assume the existence of a pseudo spectral gap [41, Section 3.1]. The bounded Radon-Nikodym derivative is a classical assumption and has been used extensively in the literature on Markov chain Monte Carlo methods [17] (lines 177-178 in our paper). As even the standard assumptions (e.g., uniform ergodicity) are difficult to verify, we believe that moving to weaker assumptions  represents a step toward conditions that practitioners can realistically justify.
>
> > The practical implications for deployment in real FL systems are not deeply explored. For example, how to estimate spectral gap or choose $K$ in practice is left unaddressed.
>
> - In the case of finite state space Markov chain, the pseudo spectral gap and $C_{\infty}$ can be computed by estimating the transition matrix and the stationary distribution. This can be done as in [47], by keeping track of the transition between different states.
>
> > The experimental section is somewhat limited, only one real dataset is used, and synthetic settings are only briefly mentioned.
>
> - We note that the main focus of our paper is on the theoretical guarantees of FL algorithms with Markovian Data. We believe that providing the first convergence results for these FL algorithms under Markovian data streams, showing that collaboration is still beneficial in this setting is a major contribution of our paper. Additional experiments with synthetic data are provided in Appendix B.7, where we also show the effect of mixing time on the convergence. These experiments also support our theoretical findings.
>
> ### Questions
>
> > The paper evaluates performance on a single real-world dataset (air-quality time series). How generalizable are the findings across domains?
>
> - Our analysis focuses on problems that can be modelled by Markov chains. The choice of the air-quality dataset used in our experiments is motivated by the fact that many physical processes exhibit Markov structure. Our results are not restricted to this specific domain, but rather apply broadly to settings where data can be modeled as Markov chains—such as reinforcement learning, financial forecasting, and sensor networks (see lines 141-144 in our paper).
>
> > You suggest that momentum-based methods (SGD-M) mitigate heterogeneity without requiring control variates (e.g., SCAFFOLD). Could you provide direct empirical comparison with control variate-based methods, such as SCAFFOLD or SCAFFLSA, under the Markovian setting?
>
> > The theoretical results require careful tuning of parameters like step sizes $\gamma$, $\eta$, and momentum $\beta$ based on unknown quantities. Is there an adaptive strategy or heuristic the authors can recommend for real-world deployment?
>
> > All your evaluation metrics focus on gradient norm as the convergence criterion. Can you report test accuracy or loss (e.g., RMSE on PM2.5) to give a sense of practical performance?
>
> - We carried out additional experiments for the air-quality time series dataset used in Section 5, we added a comparison to SCAFFOLD. To tune $\gamma$, $\eta$ and $\beta$, we use grid search over a logarithmic grid from $0.1$ to $10^{-4}$ for the learning rates and from $0.1$ to $0.9$ for the momentum coefficient. We report the *RMSE on PM2.5* averaged over 5 runs, along with the 95% confidence interval at the $80$-th communication rounds in the table below:
>
>     | Algorithm     | $K=10$            | $K=50$            | $K=100$           |
>     | ------------- | ----------------- | ----------------- | ----------------- |
>     | Minibatch SGD | $0.3997 ± 0.0054$ | $0.3774 ± 0.0012$ | $0.3776 ± 0.0005$ |
>     | Local SGD     | $0.3880 ± 0.0088$ | $0.3786 ± 0.0004$ | $0.3776 ± 0.0001$ |
>     | SCAFFOLD      | $0.3879 ± 0.0088$ | $0.3788 ± 0.0004$ | $0.3778 ± 0.0001$ |
>     | Local SGD-M   | $0.3835 ± 0.0027$ | $0.3774 ± 0.0004$ | $0.3767 ± 0.0008$ |
>
>
> - We can observe that Local SGD-M consistently achieves the lowest RMSE, while SCAFFOLD behaves quite similarly to Local SGD. We argue that the benefit of SCAFFOLD is more clear in the partial participation setting. [9] also shows that incorporating momentum into SCAFFOLD under the partial client participation setting further improves the convergence. Extending our current analysis to evaluate the combined effect of momentum and control variates in the partial participation setting with Markovian data is a promising direction for future research.
>
> - In real world deployment, where grid search is impractical, another alternative is to use adaptive step-size method. In the centralized setting, it has been shown that AdaGrad step size can adapt to the unknown mixing time of the underlying Markov chain [13]. We believe that exploring similar adaptive learning rate strategies in the federated setting is a valuable direction for future work.
>
> #### References
>
> [41] Daniel Paulin. Concentration inequalities for Markov chains by Marton couplings and spectral methods. 2018
>
> [17] Jianqing Fan, Bai Jiang, and Qiang Sun. Hoeffding’s Inequality for General Markov Chains and Its Applications to Statistical Learning. JMLR 22, 2021.
>
> [47] Angelo Rodio, Francescomaria Faticanti, Othmane Marfoq, Giovanni Neglia, Emilio Leonardi. Federated learning under heterogeneous and correlated client availability. IEEE INFOCOM 2023.
>
> [9] Ziheng Cheng, Xinmeng Huang, Pengfei Wu, and Kun Yuan. Momentum benefits non-iid federated learning simply and provably. ICLR 2024

---

> ### Comment · Reviewer_iDzF · 2025-08-05
>
> Thank you for the detailed response and for enhancing the theoretical soundness of the paper through additional experimental evaluations. As the current rebuttal has largely addressed my primary concerns, I have decided to maintain my accept recommendation.

---

> > ### Author Response · Authors · 2025-08-05
> > **Response to reviewer's comment**
> >
> > We sincerely thank the Reviewer for taking the time to read our rebuttal. We're pleased to hear that our response has addressed your concerns.

---

### Note · Authors · 2025-08-12

- We thank the Reviewers and the AC for their careful consideration of our paper and their useful suggestions. Below, we summarize the key points raised by the Reviewers.

## Non-vanishing variance term
- Reviewer T7vf and Reviewer hK1v were concerned with a non-vanishing "variance" term in our bounds. In our rebuttal, which the Reviewers agreed addressed their concerns, we highlighted that this term arises due to bias from Markovian gradient estimates, which cannot be controlled by the step-size (unlike in the i.i.d. case). In prior works [3,12,13,15] that considered centralized or token-passing settings, this bias term can vanish via introducing and tuning auxiliary parameters, but their method are unsuitable for the FL setting. Specifically,
    - [12, 15] require a minimum number of iterations (analogous to our communication rounds $T$), conflicting with FL's aim of reducing communication by leveraging local computation. In contrast, we show that increasing local computation allows FL with Markovian sampling to match the i.i.d. communication complexity lower bound.
    - [3, 13] use minibatch gradient estimates, where the batch size scales inversely with the step size or with $\sqrt{T}$. We note that this is similar to scaling $K$ in our bounds accordingly. However, we decide to choose the number of local steps $K$ as an independent parameter, which can be influenced by practical constraints such as the data arrival rate or the buffer size of client devices.

- We will provide more discussions about the limitations of existing works in Markovian SGD, further clarifying the motivation for our approach in a revised version of the paper.

## Additional experiments
- During the discussions, Reviewer iDzF and Reviewer T7vf asked for additional experiments. In response:
    - We provided additional experiments with tuned step sizes and momentum coefficient (via grid search) for the algorithms in our paper as well as SCAFFOLD. These experiments showed that SCAFFOLD is outperformed by Local SGD with momentum on the real air quality dataset considered in our paper;
    - We investigated scaling the local step size for Local SGD by $K$ (number of local steps). These experiments provided additional evidence for the tightness of our bounds by showing that the variance term cannot be controlled with a small step size.

- The Reviewers both agreed that our new experiments addressed their concerns. We will incorporate these results in a revised version of the paper.

---

### Decision · Program_Chairs · 2025-09-17

**Decision:**

Accept (poster)

**Comment:**

This paper investigates federated learning (FL) with Markov data streams. In particular, minibatch SGD, local SGD, and local SGD with momentum on Markovian data are considered. The main strength lies in its rigorous theoretical contributions and clear exposition. The main weakness about the non-vanishing variance term and the lack of experiment were fully addressed in the rebuttal. Therefore, I recommend acceptance.